# Cell-specific responses of *Anopheles gambiae* fat body to blood feeding and infection at single-nuclei resolution

Stephanie Serafim de Carvalho [1], Colton McNinch[2], Ana-Beatriz F. Barletta [1] & Carolina Barillas-Mury [1] ✉

The mosquito fat body plays key roles in metabolism and immunity, yet its cellular diversity and specialization remain poorly understood. This study analyzed 97,650 nuclei from the female *Anopheles gambiae* abdominal body wall at single-nucleus resolution, identifying seven major cell types. Fat body trophocytes are most abundant ( ̃ 85%), with five subpopulations: basal (T1, T2), metabolically enriched (T3), immune-responsive (T4), and a vitellogenic group (T5) found only in blood-fed females. Sessile hemocytes comprise 7.4% of cells and expression of lipid biosynthesis enzymes increase in oenocytes (1.1%) after immune priming. T4 trophocytes consistently express immune genes, while various cell types respond to bacterial infection. Blood feeding induces extensive transcriptomic changes, notably upregulating vitellogenin and DNA replication genes, indicating trophocyte endoreplication and metabolic shifts. Vitellogenin mRNA was expressed in the first layer of trophocytes facing the hemolymph with apical subcellular localization. This high-resolution atlas reveals specialized trophocyte roles in mosquito immunity and reproduction.

Mosquito-borne diseases continue to impose a significant global health burden, with *Anopheles gambiae* as the primary African vector of *Plasmodium falciparum* malaria, the most severe form of the disease in humans. Female mosquitoes become infected when they ingest blood from an infected person containing gametocytes. The parasite undergoes a series of developmental stages in the vector, it multiplies, and is transmitted when the mosquito feeds on a new host. Blood feeding is an important physiological event in mosquitoes, essential for reproduction and egg development, and triggers activation of complex hormonal and nutrient-derived signaling pathways that regulate blood digestion in the midgut, yolk protein precursor (YPP) synthesis in the fat body during vitellogenesis, and nutrient uptake by developing oocytes in the ovaries[1,2].

The mosquito fat body is a multifunctional organ analogous to both the adipose tissue and liver in mammals. It is the primary site of nutrient storage and energy metabolism[3], contributes to the systemic immune response by producing antimicrobial peptides and participating in pathogen melanization[4], facilitates detoxification of metabolic waste and xenobiotics[5,6], and is a major site of hemolymph protein synthesis[3]. It also plays a central role in vitellogenesis by synthesizing and secreting yolk proteins into the hemolymph that are transported to the ovaries to support egg maturation[1]. Structurally, the abdominal fat body consists of flattened lobes loosely attached to the abdominal integument that project towards the visceral organs and are in direct contact with the hemolymph[7,8]. The integument, the outermost layer of the mosquito body, is formed by an outer layer of cuticle that is synthesized by a single layer of epidermal cells supported by a basal lamina[9]. The fat body lobes vary in size and are permeated by the respiratory system, which consists of a network of tracheoles[9]. Based on their anatomical position, the abdominal fat body can be described as ventral, lateral, or dorsal.

[1]Laboratory of Malaria and Vector Research, National Institutes of Allergy and Infectious Diseases, National Institutes of Health, Rockville, MD, USA. [2]Bioinformatics and Computational Biosciences Branch, Office of Cyber Infrastructure and Computational Biology, National Institute of Allergy and Infectious Diseases, National Institutes of Health, Bethesda, MD, USA. ✉e-mail: cbarillas@niaid.nih.gov

The abdominal body wall contains the fat body and other associated cells. Trophocytes are the predominant cell type within the fat body[3]. These large polyploid cells form the fat body lobes and accumulate nutrient reserves in the cytoplasm in the form of lipid droplets and glycogen granules[7,8,10–12]. Oenocytes are rounded cells derived from ectodermal tissue, that can be individual or organized in clusters[7,13] and contribute to hydrocarbon synthesis for the cuticle[14], detoxification[15,16], sex pheromone production[13,17], and immune priming[18]. Immune priming in *An. gambiae* is mediated by the systemic release of a hemocyte differentiation factor (HDF) that consists of a complex formed by lipoxin A4 bound to Evokin, a lipid carrier[18–20]. This response is triggered when the microbiota comes in contact with the midgut epithelia during *Plasmodium* ookinete invasion of the midgut[20,21]. HDF release increases the proportion of circulating granulocytes and enhances the immune response to subsequent infections[19–21]. Oenocytes express the double peroxidase enzyme (DBLOX), which mediates HDF synthesis, and their numbers increase in primed mosquitoes[18].

Other associated fat body cells include pericardial cells, hemocytes, and neurons. Pericardial cells (PCs) are located in the dorsal body wall along the mosquito heart[9]. They flank each ostium, a strategic site of high flow as hemolymph enters the heart to be pumped out and distributed throughout the mosquito hemocoel[9,22,23]. PCs are immunocompetent cells that sense pathogens in the hemolymph and are also involved in osmoregulation and detoxification[9,22,24,25]. Hemocytes are immune cells that circulate in the hemolymph but are also often found associated with the surface of different tissues, including

the fat body. For example, periosteal hemocytes aggregate on the heart surface and can phagocytize pathogens present in the hemolymph[22]. Mosquitoes have a ventral neural cord spanning the thorax and abdomen, as part of the nervous system that coordinates sensory and locomotor signals between the brain and the body[9].

Despite the central role of the fat body and its associated tissues in mosquito physiology and immunity, the cellular composition and cell-type-specific responses to blood feeding and infection remain poorly defined. In this work, we used bulk and single-nucleus RNA sequencing (snRNA-seq) to systematically characterize the abdominal fat body of *An. gambiae* adult females. By integrating molecular profiling with morphological analyses, we identified and classified the major cell types within the fat body and adjacent tissues. Furthermore, we investigated how these cell populations transcriptionally respond to physiological stimuli such as blood feeding, as well as bacterial and *Plasmodium* infection. Our findings provide a comprehensive cell atlas of the mosquito fat body and uncover dynamic transcriptional programs that underlie its diverse functional roles in metabolism, reproduction, and immunity.

## Results
### snRNA-seq reveals 12 clusters in the abdominal body wall of *An. gambiae*

In this study, bulk and snRNA-seq were used to identify both broad transcriptome changes, as well as cell-specific responses of *Anopheles gambiae* adult female fat body trophocytes to blood-feeding, bacterial challenge and immune priming (Fig. 1A). Bulk RNA-seq was used to

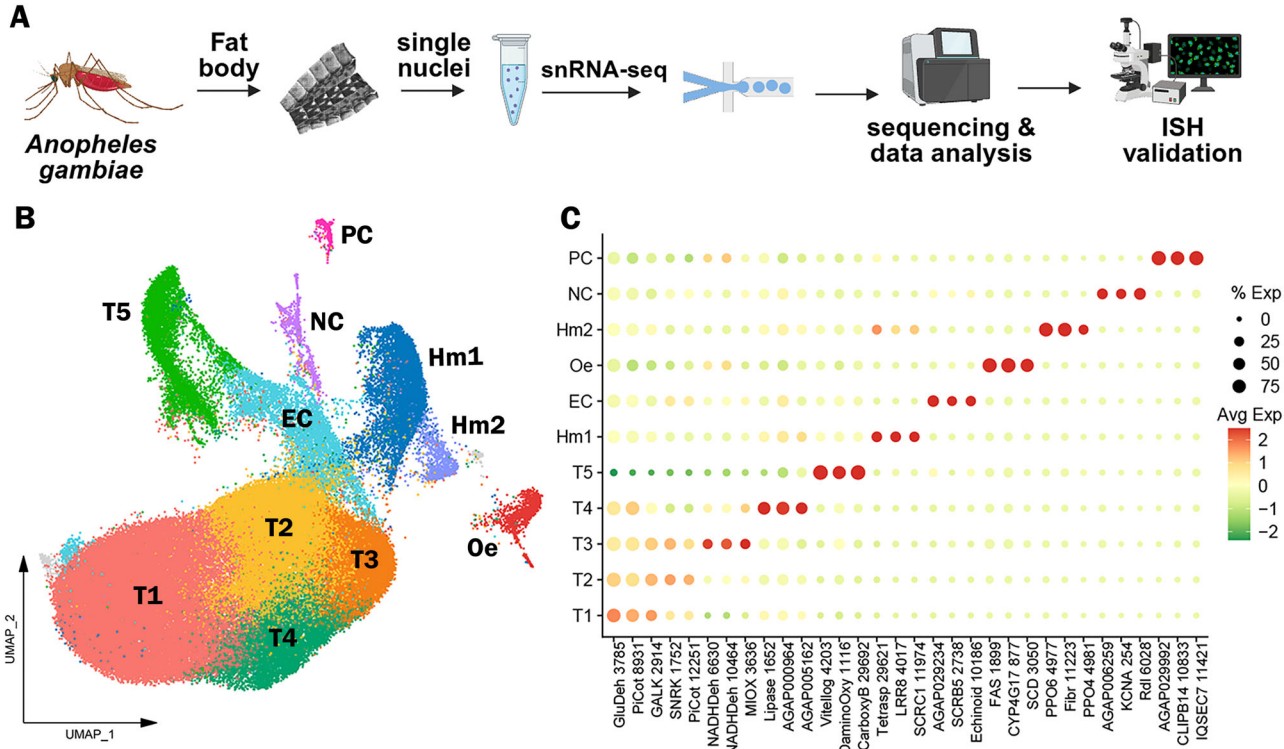

**Fig. 1 | Molecular identification of *Anopheles gambiae* fat body cell types.** **A** Schematic overview of the experimental workflow for snRNA-seq. Created in BioRender. Serafim de Carvalho, S. (2026) https://BioRender.com/qgu8rhk. **B** UMAP (Uniform Manifold Approximation and Projection) visualization of integrated data representing 97,650 nuclei isolated from *An. gambiae* body wall, colored by cluster identity. **C** Dot plot displaying two to three representative marker genes per cluster. Dot size indicates the percentage of cells (% Exp) within each cluster expressing the gene, while dot color represents the average expression (Avg Exp) level. Gene names were abbreviated, with numerical identifiers corresponding to VectorBase AGAP accession numbers. Cell type annotations: T1–T5 Trophocytes, Oe oenocytes, Hm1 and Hm2 Hemocytes, EC Epithelial cells, NC Neural cord, PC Pericardial cells. GluDeh

glucose dehydrogenase, PiCot inorganic phosphate cotransporter, GALK galactokinase, SNRK SNF related kinase, NADHDeh NADH dehydrogenase, MIOX inositol oxygenase, Vitellog Vitellogenin, DaminoOxy D-amino-acid oxidase, CarboxyB carboxypeptidase B, Tetrasp Tetraspanin, LRR8 Leucin-rich-repeat 8, SCRC1 Class C Scavenger Receptor, SCRB5 Class B Scavenger Receptor, FAS Fatty acid synthase, CYP4G17 cytochrome P450, SCD stearoyl-CoA desaturase, PPO6 prophenoloxidase 6, Fibr Fibrinogen C-terminal domain-containing protein, PPO4 prophenoloxidase 4, KCNA potassium voltage-gated channel Shaker-related subfamily A, Rdl GABA-gated chloride channel subunit, CLIPB14 CLIP-domain serine protease, IQSEQC7 IQ motif and SEC7 domain-containing protein. One biological replicate (pools of 25 mosquitoes) was performed for each condition.

capture broad transcriptional changes with high depth coverage, allowing us to quantify global differences in gene expression across different physiological and immune conditions. In contrast, snRNA-seq enabled the resolution of cell-type–specific transcriptional responses and the identification of different trophocytes subpopulations. Using both methods allowed us to distinguish tissue-wide shifts from changes restricted to specific fat body cell types, providing a more comprehensive understanding of how these stimuli reshape fat body biology.

Due to the importance of the fat body tissue on reproduction and immunity, samples were collected from the abdominal wall of healthy *An. gambiae* females fed on sugar or blood, as well as from females systemically challenged with bacteria or primed by infecting them with *Plasmodium berghei* (Fig. 1A). Samples from females challenged systemically with a mixture of *Escherichia coli* and *Micrococcus luteus* were analyzed 4 h post-injection, while the priming response was evaluated six days after females were fed on *P. berghei*-infected mice. The response of fat body cells to blood feeding was investigated 24 h after ingestion of a blood meal. The selected conditions represent the major physiological and immune challenges experienced by adult female *An. gambiae*: nutrient acquisition (sugar feeding), reproductive activation (blood feeding), acute antibacterial defense (systemic bacterial challenge), and long-term immune modulation (*Plasmodium*-mediated priming). Together, these treatments capture a broad spectrum of stimuli that shape fat body function in nutrition and vector–pathogen interactions.

A robust single-nucleus isolation protocol was established, yielding 700–1200 nuclei per microliter. The 10× Chromium technology was employed to perform 3' snRNA-seq on pooled nuclei samples. A total of 97,650 high-quality individual fat body nuclei were obtained from all samples combined. More than $2 \times 10^9$ total sequence reads from the fat body nuclei were acquired, with a median of 949 transcripts and 560 genes per nucleus, and a transcriptome mapping ratio ranging from 57.6% to 76.6%. To ensure data quality, cutoffs for the number of counts and features, as well as the percentage of mitochondrial and ribosomal content, were set for each sample (Supplementary Fig. 1). The datasets were integrated after filtering, resulting in 12 unsupervised clusters visualized using a Uniform Manifold Approximation and Projection (UMAP) (Fig. 1B). Specific gene markers for each cluster were identified (Avg log2FC > 0.5, P val adj <0.05), and candidate markers were selected for validation by in situ hybridization (ISH) (Supplementary data 1, Figs. 1C and 2). One cluster had fewer than 300 cells after integrating all samples (colored in gray in the UMAP, Fig. 1B) and could not be validated through ISH; therefore, it was not considered a relevant cell cluster and will not be further discussed.

### Identification and validation of the abdominal body wall cell types

Based on gene markers from each cluster and experimental validation through ISH, we identified eleven clusters corresponding to fat body trophocytes and other cells associated with the abdominal wall. We identified a total of seven different cell types associated with the abdominal body wall of female mosquitoes: fat body trophocytes with five subpopulations (T1, T2, T3, T4, T5), two types of hemocytes (Hm1, Hm2), epidermal cells (EC), oenocytes (Oe), neurons (NC), and pericardial cells (PC) (Figs. 1B, C, 2, Supplementary data 1).

Trophocytes were the most abundant cell type, comprising 85.2% of the cells in the dataset (Supplementary data 2). They are large polymorphic cells with cytoplasm filled with lipid droplets, organized in multiple layers that form the fat body lobes associated with the different segments of the abdominal body wall (Fig. 2A, B). Five distinct trophocyte transcriptional profiles were observed, with clusters T1, T2, T3, and T4 present in sugar-fed samples and cluster T5 present only in blood-fed samples. Galactokinase (*GALK - AGAP002914*) is a general marker gene expressed in sugar-fed trophocyte clusters (Figs. 1C and 2B). Hemocytes (Hm1, Hm2) are the second most abundant cell type, representing 7.4% of the cells (Supplementary data 2). Some are associated with the hemolymph surface of the fat body, while others permeate the trophocyte layers within the lobes (Fig. 2A, Supplementary Fig. 2). Hemocytes of the granulocyte (Hm1) lineage were the most prevalent (86.9% of hemocytes) (Fig. 2C, Supplementary Fig. 2), identified using Leucine-rich-repeat protein 8 (*LRR8 - AGAP004017*) as a marker, while Prophenoloxidase 4 (*PPO4 - AGAP004981*) expression identified oenocytoids (Hm2), representing 13.1% of hemocytes (Supplementary Fig. 2). Epidermal cells (EC) are closely associated with the cuticle (Fig. 2A, D), have small lipid droplets in their cytoplasm, and are organized in stripes (Fig. 2D). They represent 3.26% of the isolated cells (Supplementary data 2), and expression of Echinoid (*AGAP010186*) confirmed their identity in ISH.

The abdominal wall of adult females has three main regions: ventral, lateral, and dorsal, with significant differences in cell types associated with these cuticular regions (Fig. 2A). Oenocytes (Oe) are present in the ventral and lateral regions, either as individual cells, or organized as cell clusters embedded in the abdominal fat body that lack lipid droplets in their cytoplasm (Fig. 2A, E; Supplementary Fig. 3A, B). They correspond to 1.12% of the cells (Supplementary data 2), with fatty acid synthase (*FAS - AGAP001899*) as a specific gene marker (Figs. 1C and 2E). Neurons from the neural cord (NC) were observed in the ventral region (Fig. 2A, F), representing 0.84% of the cells, and expression of the resistance to Dieldrin gene (*Rdl - AGAP006028*), a GABA-gated chloride channel subunit, confirmed their identity by ISH (Figs. 1C and 2F). Finally, pericardial cells (PC) (Fig. 2G) are present in the dorsal region along the heart, representing 0.41% of cells, and express the Clip-domain Serine Protease B14 (*CLIPB14 - AGAP010833*) as a specific marker (Figs. 1C and 2A). The morphology and transcriptional characterization of the different cell types associated with the abdominal wall revealed a complex organization in the lateral, ventral, and dorsal regions.

### Oenocytes are the main cell type involved in immune priming

We performed bulk and snRNA-seq on mosquitoes challenged with *P. berghei* to assess the transcriptional and cell-specific responses to immune priming in cells associated with the mosquito body wall. Bulk RNA-seq revealed 45 upregulated genes and 1 downregulated gene (logFC > 0.5, FDR < 0.05) in the body wall of primed mosquitoes (Fig. 3A, Supplementary data 3) six days after *Plasmodium* infection. Notably, 42.2% of upregulated genes were markers of the oenocyte cluster (Fig. 3A, genes with asterisk), with 37.7% previously reported as expressed or enriched in oenocytes[14,15]. Most of these genes are related to fatty acid metabolism, including propionyl-CoA synthase (*AGAP001473*), fatty acid synthases (*FAS - AGAP001899, AGAP008468, AGAP028049*), fatty acid elongases (*AGAP005512, AGAP007264, AGAP003195, AGAP003197*), fatty acyl-CoA reductases (*FAR2 - AGAP004784, AGAP005986, AGAP005984, AGAP005985*), stearoyl-CoA desaturases (*SCD - AGAP003050, AGAP003051*), and cytochrome P450 enzyme (*CYP4G17 - AGAP000877*).

Fatty acid synthase (*FAS, AGAP001899*) mRNA expression has strong and specific expression in the oenocyte clusters present in the ventral and lateral abdominal body wall (Fig. 2E, Supplementary Fig. 3A) but is absent in the dorsal region (Supplementary Fig. 3B). We also stained the abdominal wall tissue with a specific probe to detect *DBLOX* (*AGAP008350*), an enzyme involved in lipoxin A4 synthesis, confirming the identity of oenocytes as *FAS*+ and *DBLOX*+ cells that lack lipid droplets and are individual or organized in clusters (Fig. 3B, Supplementary Fig. 3C–F). Furthermore, we confirmed by ISH that *FAS* expression induction in response to priming occurs specifically in oenocytes present in the lateral and ventral regions of the abdominal wall (Fig. 3C, D, Supplementary Fig. 3C–F, Supplementary Fig. 4). The induction in expression of *FAS* observed in bulk RNA-seq was also

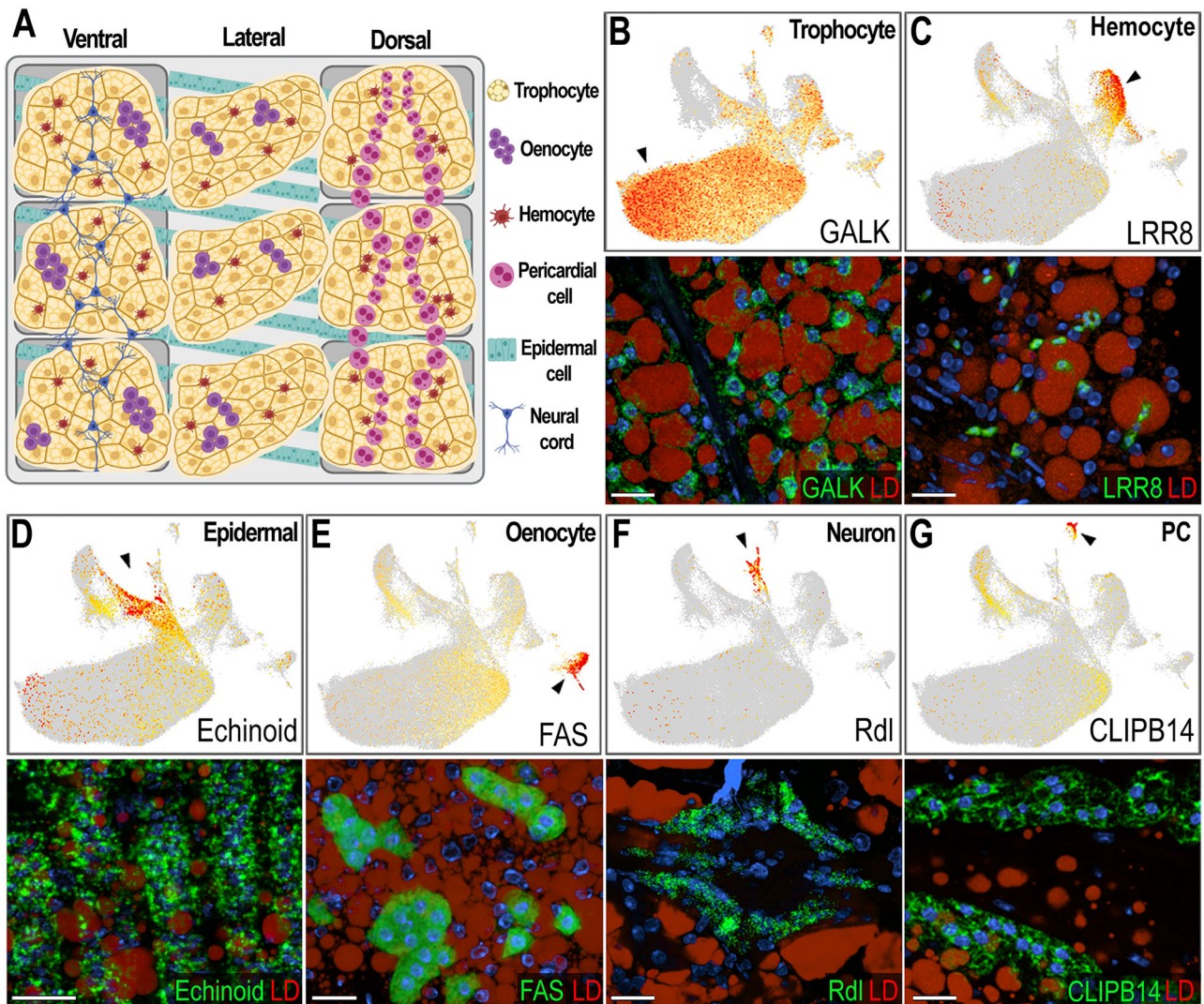

**Fig. 2 | Validation and characterization of abdominal fat body cell types in female *Anopheles gambiae*. A** Schematic illustrating the morphology and spatial distribution of abdominal fat body cell types. Gray boxes represent three segments of the abdominal wall. Created in BioRender. Serafim de Carvalho, S. (2026) https://BioRender.com/ovwpy85. **B**–**G**, top Feature plots showing expression levels of marker genes for each cell type. Cells with log-normalized expression below 1 were shown in gray. Expression levels are shown in a yellow-to-orange gradient. **B**–**G**, bottom RNA in situ hybridization in sugar-fed females confirming marker gene expression in corresponding cell types (*N* = 2, 8–10 tissues). Marker transcripts are shown in green, lipid droplets (LD) in red, and nuclei in blue. Scale bars: 20 μm. **B** Galactokinase (*GALK*; *AGAP002914*), a marker of sugar-fed trophocytes – clusters T1–T4. **C** Leucin-rich-repeat 8 (*LRR8*; *AGAP004017*), a marker of hemocytes of granulocyte lineage – cluster Hm1. **D** Echinoid (*AGAP010186*), a marker of epidermal cells – cluster EC. **E** Fatty acid synthase (*FAS*; *AGAP001899*), a marker of oenocytes – cluster Oe. **F** GABA-gated chloride channel (*Rdl*; *AGAP006028*), a marker of neurons that are part of the neural cord – cluster NC. **G** CLIP-domain serine protease B14 (*CLIPB14*; *AGAP010833*), marker of pericardial cells – cluster PC.

confirmed by ISH, which showed approximately a two-fold increase of *FAS* following priming (Supplementary Fig. 4). However, we could not detect a difference in *DBLOX* expression levels by ISH (Supplementary Fig. 3C, D). Taken together, these observations indicate that oenocytes express many key enzymes involved in lipid biosynthesis and activate their transcription in response to immune priming.

Circulating granulocytes are the final effectors of the immune priming response. To investigate the response of granulocytes associated with fat body tissue, we performed differential gene expression analysis on the granulocyte cluster (Hm1) between naïve and challenged mosquitoes. This analysis revealed 75 upregulated genes and 135 downregulated genes (Avg log2FC > 0.5, *P* val adj <0.05) (Supplementary data 4). Some upregulated genes are related to extracellular matrix and adhesion, such as chondroitin sulfate synthase (*AGAP001010*) and matrix metalloprotease 2 (*AGAP011870*). Additionally, genes related to cholesterol synthesis were upregulated, including progestin and adipoQ receptor family member 3

(*AGAP000144*), Patatin-like phospholipase domain containing 5 (*AGAP011925*), and long-chain acyl-CoA synthetase (*AGAP011603*) (Supplementary data 4). Among the downregulated genes, 18% are involved in the oxidative phosphorylation pathway, and 10% in carbohydrate metabolism (Supplementary data 4). Although we observed transcriptional differences between granulocytes from naïve and primed females, we did not observe a difference in the proportion of captured nuclei from body wall-associated granulocytes in response to immune priming.

### Functional characterization of fat body trophocyte subpopulations

Four trophocyte clusters (T1, T2, T3, and T4) were present in all sugar-fed samples analyzed, while an additional cluster (T5) appeared only in blood-fed females. In general, the transcriptional differences between trophocytes clusters are more subtle and unique markers cannot be used to identify specific subpopulations. T1 and T2 lack unique specific

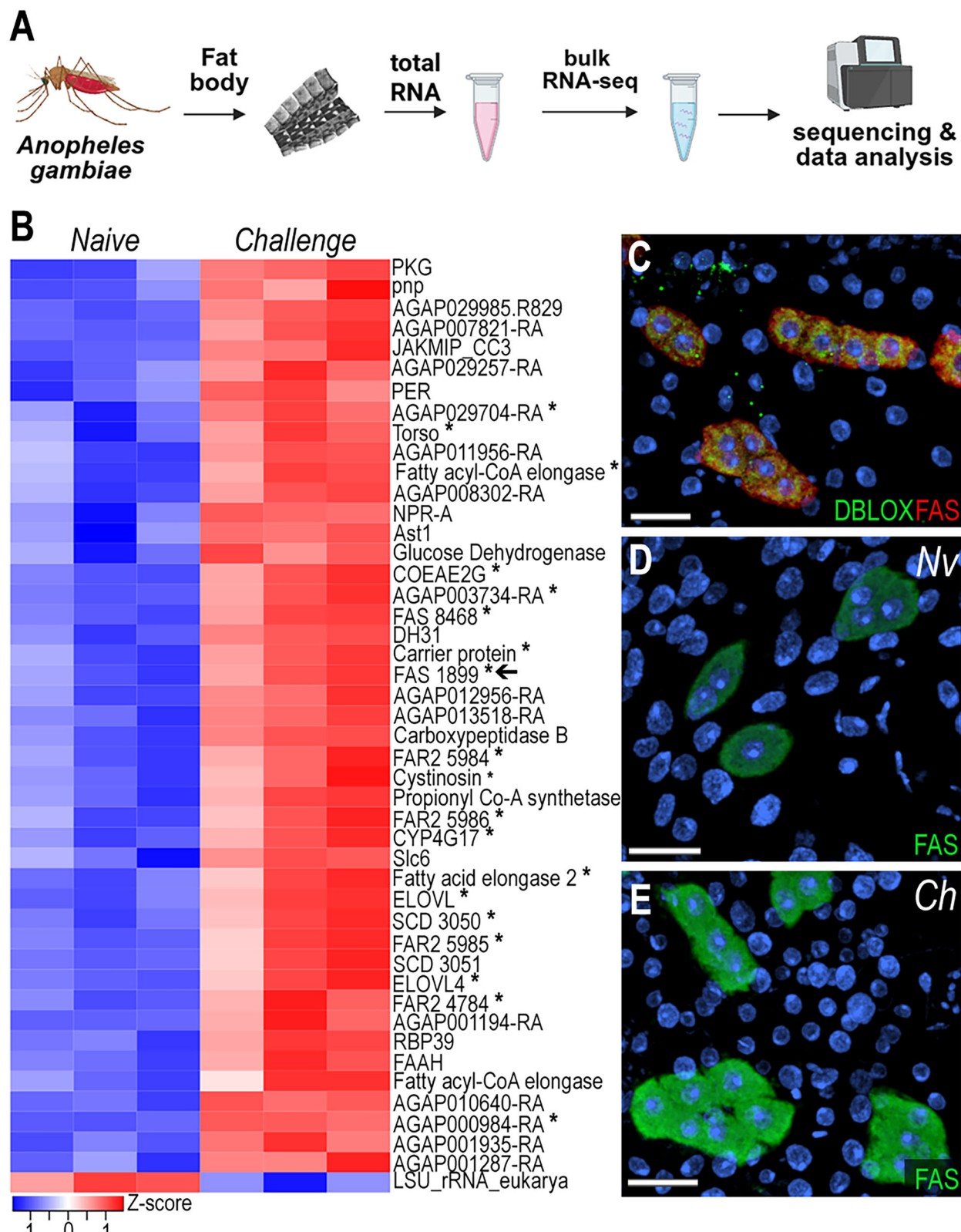

markers (Fig. 1C), whereas clusters T3, T4, and T5 express some markers at higher levels and in a higher proportion of cells than other subpopulations (Fig. 1C, Supplementary data 1). Most markers in cluster T3 are related to protein translation and metabolic pathways such as glycolysis, the tricarboxylic acid cycle, and oxidative phosphorylation (Supplementary data 1). In contrast, markers for cluster T4 are associated with immunity, melanization, or endoplasmic reticulum

(ER)-related proteins (Supplementary data 1). The expression of different functional classes of marker genes suggests that they represent distinct trophocyte subpopulations that perform specialized functions in the abdominal fat body.

To further characterize transcriptional differences between trophocyte clusters in sugar-fed mosquitoes, we performed ISH using galactokinase (*GALK - AGAP002914*) as a general trophocyte marker

**Fig. 3 | Oenocytes induce lipid synthesis genes following immune priming by *Plasmodium berghei* challenge. A** Schematic overview of the experimental workflow for bulk RNA-seq. Created in BioRender. Serafim de Carvalho, S. (2026) https://BioRender.com/ufbl2lj. **B** Heatmap of differentially expressed genes from bulk RNA-seq analysis (absolute log$_2$ fold change > 0.5 and FDR < 0.05). Asterisk (*) indicates genes that are upregulated following *P. berghei* challenge and are also markers of oenocytes ($N = 3$, 10–15 tissues). **C** RNA in situ hybridization of double peroxidase (*DBLOX; AGAP008350*) and fatty acid synthase (*FAS; AGAP001899*) in oenocytes on the lateral abdominal body wall under naïve condition ($N = 2$, 8–10 tissues). *DBLOX* is shown in green, *FAS* in red, and nuclei in blue. RNA in situ hybridization of *FAS* in oenocytes on the lateral abdominal body wall under naïve

(**D**) and *P. berghei*-challenge (**E**) conditions ($N = 2$, 10–12 tissues). *FAS* is shown in green and nuclei in blue. Scale bars: 20 μm. PKG Protein kinase cGMP-dependent, pnp purine-nucleoside phosphorylase, JAKMIP_CC3 JAKMIP_CC3 domain-containing protein, PER period circadian protein, Torso tyrosine-protein kinase receptor torso, NPR-A atrial natriuretic peptide receptor A, Ast1 allatostatin 1, COEAE2G carboxylesterase, FAS Fatty acid synthase, DH31 diuretic hormone 31, FAR2 fatty acyl-CoA reductase 2, CYP4G17 cytochrome P450, Slc6 solute carrier family 6, ELOVL Elongation of very long chain fatty acids protein, SCD stearoyl-CoA desaturase, RBP39 RNA-binding protein 39, FAAH fatty-acid amide hydrolase 2, LSU_rRNA_eukarya Eukaryotic large subunit ribosomal RNA.

with higher expression in the T1 cluster; inositol oxygenase (*MIOX* - *AGAP003636*) as a marker enriched in T3, and lipase (*AGAP001652*) in the T4 trophocyte cluster (Fig. 4A). Galactokinase was expressed in all trophocytes of sugar-fed females, with heterogeneity in the levels of expression (Fig. 4B, C). Among the *GALK+* trophocytes, some also expressed high levels of *MIOX* (*GALK$^+$, MIOX$^h$*) (Fig. 4C, D), others expressed *GALK* and high levels of the lipase (*AGAP001652*) marker (*GALK$^+$, Lipase$^h$*) (Fig. 4C, E); while a few trophocytes had strong expression of both markers (*MIOX$^h$, Lipase$^h$*) (Fig. 4F, white arrow). These findings provide direct evidence that although trophocytes have similar morphology there is heterogeneity in the combination of markers they expressed, indicative of some level of specialization.

## Immune response of the abdominal body wall to bacterial challenge

The fat body plays an important role in the mosquito systemic immune response by producing antimicrobial peptides and phenoloxidase enzymes that are secreted into the hemolymph and limit bacterial proliferation and mediate pathogen melanization, respectively. However, cell type-specific responses remain poorly understood. The transcriptional response of the abdominal body wall to bacterial infection was explored by challenging adult females with a systemic injection of a mixture of a Gram-negative (*E. coli*) and a Gram-positive bacteria (*M. luteus)*, or a sterile PBS control, and harvesting the abdominal body wall 4 h later. This approach ensured the direct delivery of a standardized bacterial load into the hemolymph, triggering a robust and physiologically relevant immune response. The combination of Gram-positive and Gram-negative bacteria provided a broad immune stimulus, ensuring the activation of the major immune signaling pathways.

Bulk RNA-seq analysis revealed 386 upregulated and 232 down-regulated genes (logFC > 1, FDR < 0.05) in response to bacterial challenge (Supplementary Fig. 5). We observed a major induction of several antimicrobial peptides such as defensin 1 (*DEF1*) and cecropins (*CEC1, CecB*), as well as multiple components of the Toll and IMD pathways, including the transcription factor REL2 and Peptidoglycan recognition proteins (*PGRPLB, PGRPLA, PGRPS1*) (Supplementary Fig. 5, Supplementary data 5). Bacterial challenge induced expression of many other immunoregulatory genes such as C-type lysozymes (*LYSC1, LYSC2, LYSC7*), clip-domain serine proteases (*CLIPA1, CLIPA2, CLIPA4, CLIPA5, CLIPA12, CLIPA14, CLIPB5, CLIPB6, CLIPB14, CLIPB15, CLIPB17, CLIPB20, CLIPB36, CLIPC7, CLIPE1*), *Anopheles Plasmodium*-responsive leucine-rich repeat proteins (*APL1A, APL1B, APL1C*), Leucine-rich repeat immune proteins (*LRIM1, LRIM4, LRIM20*), and thioester-containing proteins (*TEP1, TEP2, TEP3, TEP4, TEP6, TEP15*) (Supplementary Fig. 5, Supplementary data 5). Most downregulated genes are involved in carbohydrate, amino acid, and fatty acid metabolism, as well as membrane transport (Supplementary Fig. 5, Supplementary data 5).

To identify cell-specific responses to bacterial challenge, we compared the genes with enhanced expression after bacterial challenge in the bulk RNA-seq data with markers highly expressed in each of the cell clusters identified by snRNA-seq. Interestingly, we observed that 44 gene markers of cluster T4 were upregulated in the bulk RNA-

seq data after bacterial challenge (Fig. 5A), suggesting that T4 trophocytes mount a strong immune response to infection. To understand the responses of fat body and tissue-associated cells to infection, we performed ISH using probes for *DEF1* and *TEP1* (Fig. 5B–F). *TEP1* is a marker of cluster T4 and is only expressed in a subset of trophocytes in the PBS control (Fig. 5B, left). However, *TEP1* is expressed in all trophocytes following a bacterial challenge (Fig. 5B, right). *DEF1* is a shared marker of the trophocyte T4 and hemocyte (Hm1 & Hm2) clusters. We were unable to detect *DEF1* expression in the control samples (Fig. 5C, left). However, after bacterial challenge, *DEF1* is highly expressed in a subset of trophocytes (Fig. 5C, right), as well as in hemocytes (Fig. 5D, lower panel), epidermal cells (Fig. 5E, lower panel), and pericardial cells (Fig. 5F, lower panel).

Differential expression analysis on trophocyte subpopulations revealed a core immune response with 84 genes being upregulated and 54 genes being downregulated in all trophocyte clusters (T1–T4) (Avg log2FC > 1, P val adj <0.001) (Fig. 5G, H, Supplementary data 6). These genes with broad and strong induction included antimicrobial peptides (*DEF1, CEC1, CecB*), genes involved in melanization (*CLIPB17, CLIPC7, CLIPB15*), the complement system (*TEP1, APL1C, CLIPA2*), and other canonical immune genes, while mitochondrial (*ND6, AGAP001502, AGAP007975*) and lipid metabolism genes (*AGAP000844, AGAP008369, AGAP012352*) were downregulated (Supplementary data 6). Interestingly, each trophocyte subpopulation also had specific transcriptional responses to bacterial infection (Fig. 5G, H). As expected, T4 trophocytes had the strongest transcriptional response to infection, inducing expression of 57 genes, many of which are involved in immunity, such as pathogen recognition proteins (*GNBPB1, PGRPLA*), activation of the Toll pathway (*AGAP001798*), bacterial cell wall lysis (*LYSC2, AGAP001430*), microtubule assembly and cytoskeleton reorganization (*AGAP012991, AGAP010770, Arf2*), protein folding, maturation, and translocation (*AGAP010225, AGAP002816, AGAP007361*), and intracellular signaling (*AGAP029529, AGAP010350, AGAP005352*) (Fig. 5G, Supplementary data 6). Downregulated genes in cluster T4 are involved in cholesterol metabolism (*AGAP003542, AGAP004845*), mitochondrial translation (*AGAP009737, mRpL47*), transcription (*AGAP001106, AGAP029767*), and intracellular signaling (*AGAP002250, AGAP001110*) (Fig. 5H, Supplementary data 6). Most genes in T1 trophocytes were downregulated (29 genes), and 48% of them are involved in oxidative phosphorylation (Fig. 5H, Supplementary data 6). T2 trophocytes induced expression of 10 genes, 7 of which are related to protein folding, protein and vesicle transport, and autophagy (*APG8*) (Supplementary data 6). Finally, T3 trophocytes increased the expression of 14 genes, some of which are involved in immunity, ER stress, or membrane remodeling (Supplementary data 6). Differential expression analysis of hemocyte, epidermal cells, and pericardial cell clusters revealed upregulation of several core immune response genes that are also induced in the trophocyte clusters (Supplementary data 6).

## Response of the fat body trophocytes to blood feeding
The cellular response of trophocytes to blood feeding was investigated by comparing bulk RNA-seq and snRNA-seq of the abdominal fat body

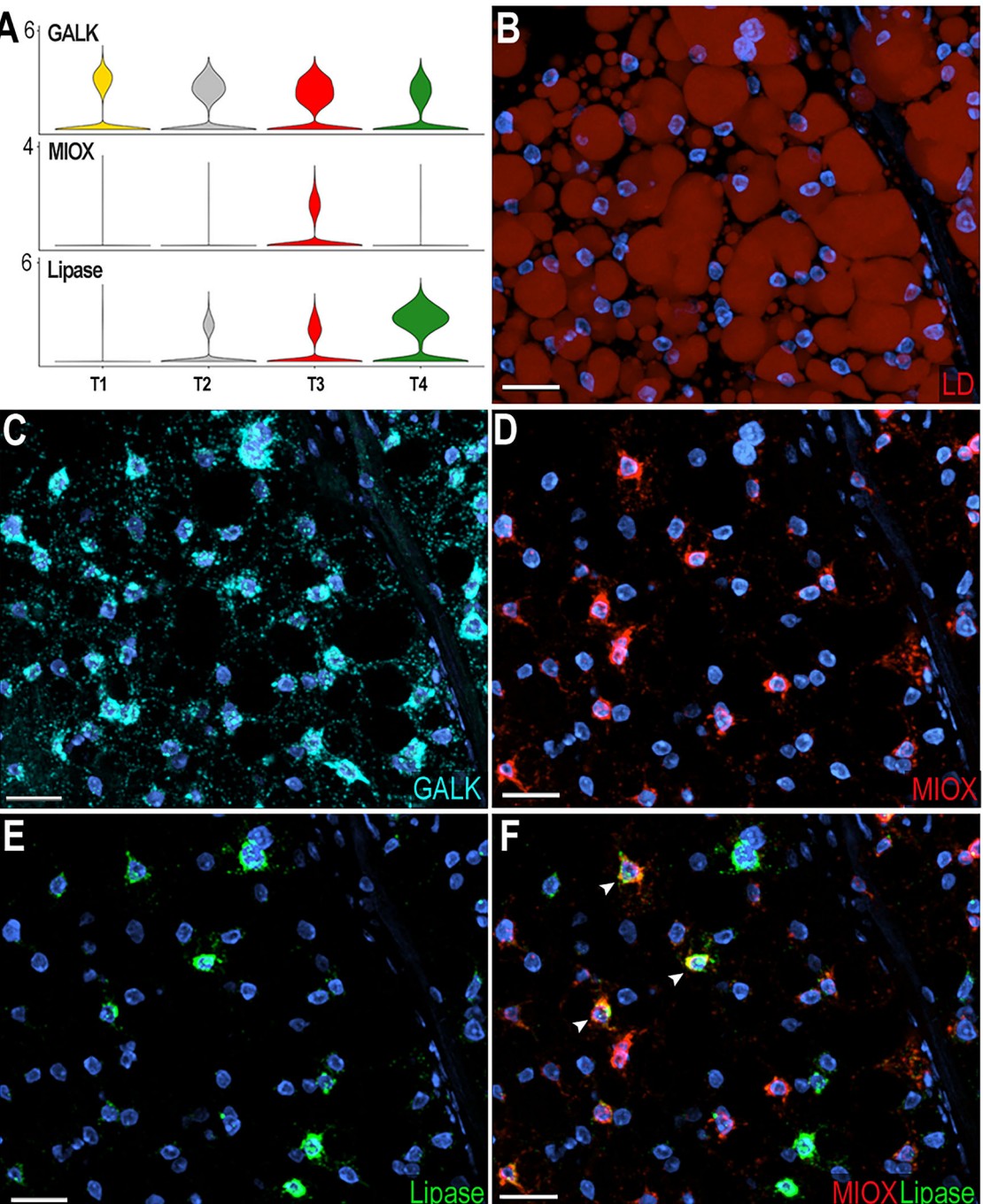

**Fig. 4 | Characterization of trophocytes subpopulations in sugar-fed mosquitoes. A** Violin plots showing expression of Galactokinase (*GALK*; *AGAP002914*), Inositol oxygenase (*MIOX*; *AGAP003636*), and Lipase (*AGAP001652*) in trophocytes. **B** Lipid staining of sugar-fed mosquito fat body showing lipid droplets (LD) distributed throughout the trophocyte cytoplasm (*N* = 2, 10–15 tissues). LD are stained in red. **C–F** RNA in situ hybridization validating trophocyte subpopulation markers

(*N* = 2, 10–15 tissues): **C** *GALK*, marker of the T1 trophocyte subpopulation. **D** *MIOX*, marker of the T3 trophocyte subpopulation. **E** Lipase, marker of T4 trophocyte subpopulation. **F** Merged image showing *MIOX* (red) and Lipase (green) expression. White arrows indicate double-positive cells. *GALK* is shown in cyan and nuclei in blue. *N* = 2, 10–15 tissues. Scale bars: 20 μm.

between sugar-fed and blood-fed females 24 h post-feeding (PF). In the sugar-fed sample, cluster T1 corresponds to 60.6% of the cells, while in the blood-fed sample (24 h PF), the same cluster represents only 5.1% of the cells. Conversely, cluster T5 represented 0.1% of cells in the sugar-fed sample and becomes 54% of the cells 24 h PF (Fig. 6A, B, Supplementary data 2). These dramatic changes in the numbers of cells in these clusters suggest that cluster T1 undergoes such a drastic transcriptional change in response to blood feeding that the integration algorithm classifies it as a new cluster (T5), which is only present in

blood-fed females. RNA velocity analysis of blood-fed samples indicated enhanced transcriptional dynamics within cluster T5, evidenced by an enrichment of unspliced transcripts and a predominance of positive velocity vectors (Supplementary Fig. 6). In agreement with this hypothesis, cluster T5 is greatly reduced and cluster T1 becomes predominant again 6 days PF (Fig. 6C, Supplementary data 2), when the gonadotrophic cycle has been completed, indicating that the transition of the T1 cluster to T5 is due to a prominent but transient transcriptional response to blood feeding.

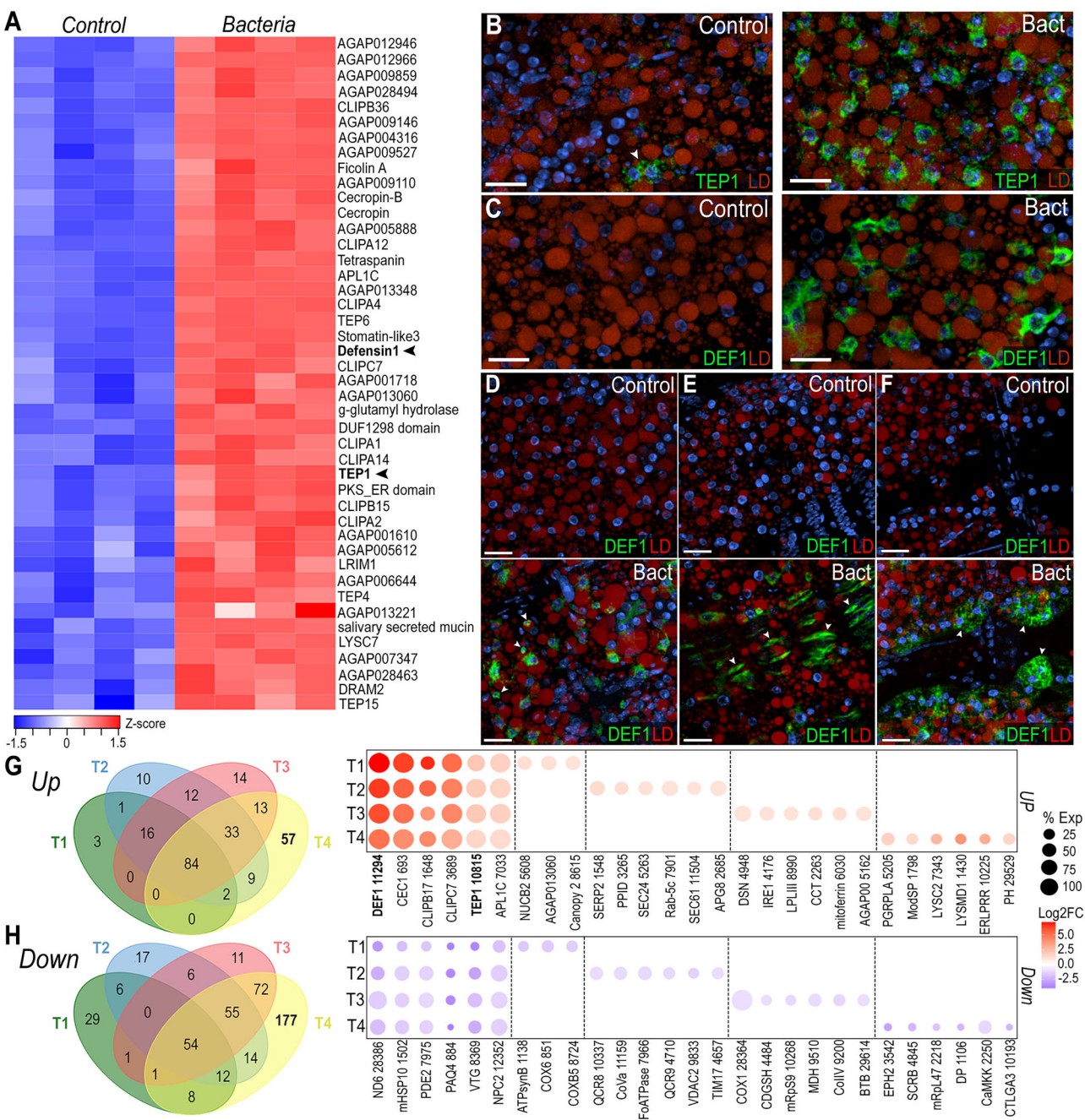

To investigate the transition of the T1 cluster to the T5 transcriptional profile, ISH of the abdominal fat body of sugar-fed and blood-fed females collected 24 h and 6 days post-feeding was performed. *GALK* was used as a general marker of sugar-fed trophocytes, with an enriched expression in the T1 cluster and vitellogenin (*Vg* – *AGAP004203*) as a marker of the T5 cluster. In sugar-fed females, *GALK* is expressed in the cytoplasm of all trophocytes, particularly in the perinuclear region, while *Vg* expression could not be detected (Fig. 6D, E). In contrast, high levels of *Vg* expression were detected 24 h PF, while *GALK* expression levels were low (Fig. 6D, E). *Vg* is expressed at higher levels in the fat body associated with the lateral cuticle, relative to the ventral and dorsal sides of the tissue (Supplementary Fig. 7). Interestingly, *Vg* mRNA is only detected in the layer of fat body trophocytes in contact with the hemolymph (Fig. 6E, F). Furthermore, at a subcellular level, *Vg* mRNA is concentrated on the apical surface of the cell that faces the hemolymph (Fig. 6F). The *Vg* mRNA subcellular localization was similar in the lateral, ventral, and dorsal fat body (Supplementary Fig. 7). As expected, *GALK*

and *Vg* expression 6 days PF are similar to those of sugar-fed females (Fig. 6D, E).

Differential gene expression analysis between clusters T1 (sugar-fed) and T5 (24 h PF) revealed that 171 genes were upregulated and 341 downregulated (Avg log2FC > 1, *P* val adj <0.05) (Supplementary data 7). During vitellogenesis, the fat body produces yolk protein precursors that are transported to the ovaries. Vitellogenins (*AGAP004203*, *AGAP013109*), the main yolk proteins, are highly increased in cluster T5 (Supplementary data 7). Also, proteins involved in breaking down yolk proteins to ensure nutrient availability for the developing oocyte, such as cathepsin B precursors (*AGAP004531*, *AGAP004534*), and vitellogenic carboxypeptidase-like protein (*AGAP005434*) were upregulated in cluster T5 (Supplementary data 7). Lipids are also transported from the fat body to the developing oocytes, and several proteins involved in lipid metabolism and transport, including lipophorin (*AGAP001826*), the main hemolymph lipid transporter, were induced in cluster T5 (Supplementary data 7). In

**Fig. 5 | The T4 trophocyte subpopulation is activated following systemic bacterial infection. A** Heatmap of differentially expressed genes from bulk RNA-seq analysis (absolute log$_2$ fold change >1 and FDR < 0.05) that are also markers of the T4 cluster. Black arrows indicate Defensin 1 (*DEF1*; *AGAP011294*) and thioester-containing protein 1 (*TEP1*; *AGAP010815*) (*N* = 4, 10–15 tissues). RNA in situ hybridization of *TEP1* (**B**) and *DEF1* (**C**) in trophocytes under control (left) and systemic bacterial infection (Bact, right) conditions (*N* = 2, 10–15 tissues). RNA in situ hybridization of *DEF1* in hemocytes (**D**), epidermal cells (**E**), and pericardial cells (**F**) under control (top) and systemic bacterial infection (bottom) conditions. *TEP1* and *DEF1* are shown in green, lipid droplet (LD) in red, and nuclei in blue. White arrows indicate the corresponding cell types. Scale bars: 20 μm. Venn diagram and dot plot showing upregulated (**G**) and downregulated (**H**) genes in trophocyte subpopulations T1–T4 (average log$_2$ fold change >1 or <−1 and adjusted *P* value < 0.05). Dot size indicates the percentage of cells (% Exp) within each cluster expressing the gene, while dot color represents the log$_2$ fold change (Log2FC). Gene names were abbreviated, with numerical identifiers corresponding to VectorBase AGAP accession numbers. CLIP CLIP-domain serine protease, APLC1 Anopheles Plasmodium-responsive Leucine-Rich Repeat 1C, TEP thioester-containing protein, LRIM leucine-rich immune protein, LYSC C-type lysozyme, DRAM2 DNA damage-regulated autophagy modulator protein 2, CEC1 Cecropin 1, NUCB2 nucleobindin-2, SERP2 Stress-associated endoplasmic reticulum protein 2, PPID peptidyl-prolyl isomerase D, SEC24 transport protein SEC24, Rab-5c Ras-related protein Rab-5C, SEC61 transport protein SEC61 subunit beta, APG8 autophagy related gene, DSN duplex-specific nuclease, IRE1 endoribonuclease IRE1, LPLIII lysophospholipase III, CCT choline-phosphate cytidylyltransferase, PGRP peptidoglycan recognition protein, ModSP modular serine protease-like, LYSMSD1 lysM and putative peptidoglycan-binding domain-containing protein 1, ERLPRR ER lumen protein retaining receptor, PH PH domain-containing protein, ND NADH dehydrogenase, mHSP10 mitochondrial 10 kDa heat shock protein, PDE2 pyruvate dehydrogenase E2 component, PAQ4 Progestin and adipoQ receptor family member 4, VTG Vitellogenin domain-containing protein, NPC2 Niemann-Pick C2 protein, ATPsynB F-type H + -transporting ATPase subunit b, COX cytochrome c oxidase, QCR ubiquinol-cytochrome c reductase, CoVa cytochrome c oxidase subunit Va, FoATPase F-type H + -transporting ATPase subunit gamma, VDAC2 voltage-dependent anion-selective channel protein 2, TIM17 mitochondrial inner membrane translocase subunit, CDGSH CDGSH iron sulfur domain-containing protein 2 homolog, mRpS9 mitochondrial 28S ribosomal protein S9, MDH malate dehydrogenase, ColIV collagen type IV alpha, BTB BTB domain-containing protein, EPHX2 Epoxide hydrolase 2, SCRB scavenger receptor class B member, mRpL47 mitochondrial 39S ribosomal protein L47, DP Disconnected protein, CaMKK calcium/calmodulin-dependent protein kinase kinase, CTLGA3 C-type lectin.

---

addition, genes related to amino acid degradation, ribosomal proteins, endoplasmic reticulum (ER) stability, protein transport, and glycosylation were increased (Supplementary data 7). Meanwhile, genes related to autophagy, carbohydrate metabolism, circadian clock, immunity, signaling, and transcription were downregulated (Supplementary data 7).

Bulk RNA-seq analysis confirmed a strong transcriptional response to blood feeding, with 1655 genes upregulated and 1368 genes downregulated (logFC > 1, FDR < 0.01) (Supplementary Fig. 8, Supplementary data 8). In agreement with the differential gene expression of cluster T5, vitellogenin (*AGAP004203* had a 5898-fold increase), cathepsin B precursor (*AGAP004534*, 121-fold increase), and expression of vitellogenic carboxypeptidase proteins (*AGAP005434*, 2.8-fold increase; *AGAP007505*, 2.8-fold increase) was also increased 24 h PF (Supplementary Fig. 8, Supplementary data 8). Interestingly, increased expression of genes involved in DNA replication, such as DNA replication licensing factors (*MCM3, MCM4, MCM5, MCM6*), DNA polymerase subunits (10 different subunits), replication factors (A1, A2, A3, and 3 C-subunits), proliferating cell nuclear antigen (*PCNA*), ribonuclease H2 subunit B, helicase Dna2, and DNA ligase 1 was observed; as well as several genes related to DNA repair (*MSH2, Rad62, RAD50, MRE11, ERCC-1*), and cell cycle such as cyclins (A, C, H, T) and cyclin-dependent kinases (*CDK1, CDK4*) (Supplementary Fig. 8, Supplementary data 8).

To determine which cells associated with the abdominal wall were undergoing DNA replication in response to blood feeding, 5-ethynyl-2′-deoxyuridine (EdU) was injected into the hemolymph of sugar-fed or blood-fed females, 6 h PF. Samples of sugar-fed females and blood-fed females were analyzed 18 h after EdU injection. We were not able to detect EdU incorporation in trophocytes of sugar-fed females (Fig. 6G). In contrast, the majority of trophocytes from blood-fed females incorporated EdU in their DNA, resulting in strong nuclear staining (Fig. 6H). A few EdU$^+$ cells with small nuclei were observed in both sugar-fed and blood-fed mosquitoes (Fig. 6G, H, white arrows). Notably, some of these small-nuclei cells were positive for the hemocyte marker *LRR8* (Supplementary Fig. 9). EdU staining was not detected in any other cell types. Staining for phosphohistone H3 (pH3), a marker of mitosis, was negative in trophocytes of sugar-fed and blood-fed females (Fig. 6G, H). Only a few pH3$^+$ hemocytes (Fig. 6H, yellow arrows) were observed in the body wall of blood-fed mosquitoes; all other cells were negative. Blood feeding also triggered a strong downregulation of genes involved in carbohydrate metabolism, such as genes involved in glycolysis, citric acid cycle, beta oxidation, elongation of very long fatty acids, ribosomal biogenesis, mitochondrial organization, protein folding, and the TOLL pathway (Supplementary Fig. 8, Supplementary data 8).

## Discussion

Single-cell transcriptome analysis has transformed our understanding of cellular heterogeneity by enabling the analysis of individual cells and has provided new insights into cell-specific functions, morphology, and dynamic transcriptional states. In this study, we used snRNA-seq to investigate the transcriptional landscape of the abdominal fat body and associated tissues of *An. gambiae* females to overcome the technical challenge of isolating intact single cells from a tissue with very high lipid content. Trophocytes are large cells with the cytoplasm filled with lipid droplets, making them fragile to high-quality single-cell isolation. Our optimized protocol involves a delipidation step, because high lipid content in the sample can lead to nuclei aggregation and interference with 10X single-cell partitioning and Gel Bead-in-Emulsion (GEM) formation. In our experience, we were unable to obtain high-quality intact fat body cells using enzymatic methods for tissue dissociation. Previous studies have demonstrated that the nuclear transcriptome correlates with the cytoplasmic mRNA composition[26,27], making snRNA-seq a viable alternative for transcriptomic profiling when whole-cell isolation is not feasible.

Previous single-cell and single-nucleus transcriptomic studies in insects have begun to illuminate the cellular complexity of metabolically and immunologically active tissues such as the fat body. In *Drosophila melanogaster*, snRNA-seq has been used to resolve cell-type-specific responses to immune challenge and aging, revealing sub-populations of fat body cells with specialized metabolic or inflammatory functions[28]. snRNA-seq was applied to *Bombyx mori* to characterize distinct metabolic and immune programs in fat body cells during viral infection, also using nuclear-based transcriptional profiling[29]. More recently, scRNA-seq and metabolomics were used to investigate the transcriptional and metabolic responses of the *Aedes aegypti* fat body and midgut following a blood meal, uncovering functionally distinct cell types and dynamic responses to blood feeding[30]. Despite these advances, a comprehensive understanding of fat body cell heterogeneity and its response to infection and immune priming has not been available for anopheline vectors of malaria. *Drosophila melanogaster* and *Bombyx mori* are non-hematophagous insects that feed continuously, and that fundamental dietary difference shapes fat body biology in ways that contrast with mosquitoes. In *Drosophila*, the fat body forms diffuse sheet-like layers with compact segmental oenocyte clusters[31], whereas *B. mori* possesses large, well-defined lobes and abundant oenocytes[32]. In mosquitoes such as *Ae.*

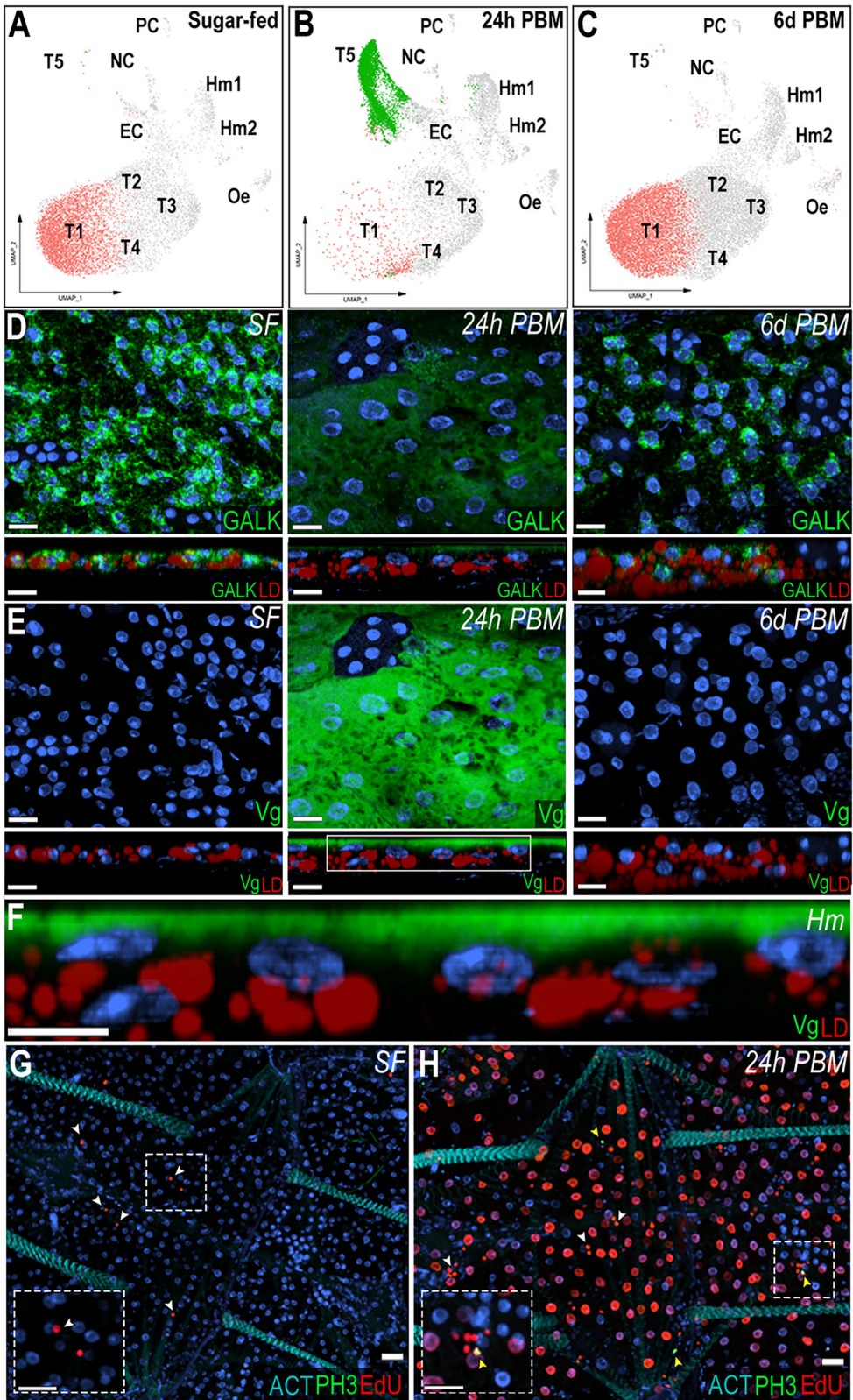

*aegypti* and *An. gambiae*, however, the requirement of a blood meal for egg development drives drastic and transient transcriptional and physiological changes in fat body trophocytes to sustain vitellogenesis[7,12]. The mosquito fat body is organized into distinct lobes with more dispersed oenocyte clusters[7]. In our study, we identified different trophocyte subpopulations, as well as oenocytes and other fat body-associated cell types, including hemocytes, epidermal

cells and neuronal cells, in agreement with previous single cell studies in insects[28–30].

Our observation of a high number of hemocytes closely associated with the fat body (Fig. 2C, Supplementary data 2) underscores the importance of local immune surveillance in this tissue. We found that while granulocytes typically represent only a small fraction (1–5%) of circulating hemocytes[20,21], they account for nearly 87% of fat body-

**Fig. 6 | Trophocytes response to blood feeding involves intense transcriptional changes.** UMAP visualization of T1 and T5 trophocyte clusters in sugar-fed (**A**), 24 h post-blood meal (PBM) (**B**), and 6 days post-blood meal (PBM) (**C**) conditions. **D, E**, top RNA in situ hybridization of Galactokinase (*GALK*; *AGAP002914*) (**D**) and Vitellogenin (*Vg*; *AGAP004203*) (**E**) in trophocytes on the lateral abdominal body wall under sugar-fed (left), 24 h post-blood meal (middle), and 6 days post-blood meal (right) conditions (N = 3, 10–15 tissues). **D, E**; bottom Side view of *GALK* and *Vg* staining. Scale bars: 15 μm. **F** Zoom-in of the selected region from *Vg* panel at 24 h post blood meal. Scale bars: 10 μm. *GALK* and *Vg* are shown in green, lipid droplet (LD) in red, and nuclei in blue. Hm indicates the hemolymph side. 5-ethynyl-2′-deoxyuridine (EdU) and phospho-histone H3 (pH3) staining in sugar-fed (**G**) and 24 post blood meal (**H**) conditions (N = 2, 10–15 tissues). EdU is shown in red, pH3 in green, Actin (ACT) in cyan, and nuclei in blue. White arrows indicate EdU-positive hemocytes and yellow arrows indicated pH3-positive hemocytes. Dashed squares show a zoom-in of hemocytes. Scale bars: 20 μm.

associated hemocytes (Supplementary Fig. 2, Supplementary data 2), suggesting that a substantial proportion of granulocytes embedded in the trophocyte layers of the tissue may represent resident immune cells. This supports the growing recognition of tissue-resident immune cells in insects that respond readily to local infections[33–35]. In contrast, granulocytes loosely attach to the hemolymph side of fat body basal lamina and circulating oenocytoids, which are more abundant in circulation and underrepresented in the fat body, may have a more systemic role and mobilize to remote sites of damage or infection.

We observed a doubling of the number of fat body-associated granulocytes following a bacterial challenge in snRNA-seq, suggesting either proliferation of resident cells and/or recruitment from circulation. This aligns with previous reports of hemocyte accumulation in the periosteal regions post-infection[35] and supports the idea that immune challenge could trigger dynamic shifts in hemocyte distribution and abundance. Additionally, the increased capture of both granulocytes and oenocytoids after a blood feeding agrees with previous reports of increased number of circulating hemocyte in blood-fed females, illustrating dynamic changes in tissue-associated hemocyte populations[36]. The presence of pH3- and EdU-positive hemocytes (Fig. 6G, H), indicating mitotic activity and DNA synthesis within these cells is indicative of cell proliferation. Together, these findings highlight the plasticity of mosquito hemocyte populations and suggest that the fat body serves not only as an immune-responsive tissue but also as a compartment for immune cell residence and activity. Future studies to explore the relative contribution of proliferation of fat body-associated hemocytes and recruitment of hemocytes circulating in the hemolymph will require longitudinal tracking and quantification of hemocyte populations in the same experimental samples, using a combination of single-cell RNA sequencing and high-resolution tissue imaging.

Oenocytes play a key role in the immune priming response. In *Aedes aegypti*, they represent ~5.9–9.8% of fat body cells in adult females[7]. We observed a strong increase in fatty acid synthase mRNA levels in oenocytes from primed females (Fig. 3C, D). We confirmed that the *DBLOX* mRNA is expressed in oenocytes, and that modest increase in *DBLOX* mRNA expression previously reported in primed females (1.5–2.0-fold) is due to the increase in oenocyte numbers[18], as we did not observe increased gene expression by ISH on individual cells from primed females. Although hemocyte numbers did not change, we do observe a distinct transcriptional response characterized by increased expression of extracellular matrix and adhesion-related genes (Supplementary data 2, 4) in primed females. This suggests enhanced hemocyte patrolling or stronger interaction with the fat body tissue. Additionally, upregulation of genes involved in long-chain fatty acid synthesis and cholesterol metabolism in hemocytes suggests that they may also synthesize bioactive lipids in response to immune priming (Supplementary data 4).

Of the five trophocyte clusters that were identified (Fig. 1B), clusters T1 and T2 shared similar transcriptomes with subtle differences in gene expression (Fig. 1C), indicative of a basal metabolic state. In contrast, clusters T3, T4, and T5 exhibited specialized gene expression patterns: T3 was enriched for metabolic genes, T4 for immune-related genes, and T5 for vitellogenesis-related genes (Supplementary data 1, Fig. 6A–C). Our findings reveal that the mosquito mounts a robust and multilayered immune response to bacterial

infection, involving not only trophocytes but also hemocytes, epidermal cells, and pericardial cells (Fig. 5B–F). The activation of both TOLL and IMD pathways, along with the induction of pattern recognition receptors, melanization components, and antimicrobial effectors, points to a broad engagement of classical innate immune mechanisms. Importantly, the identification of a core set of immune genes expressed across multiple cell types suggests a conserved, systemic response to infection. This shared transcriptional program is complemented by cell-type-specific immune signatures, indicating functional specialization that may allow for spatial or temporal modulation of the immune response. The expression of immune genes in T4 trophocytes, even in the absence of infection, raises the possibility that this subpopulation plays a role in immune surveillance or maintaining immune readiness—functions that parallel tissue-resident immune cells in vertebrates. These immune responses appear to come at a metabolic cost, as evidenced by the downregulation of oxidative phosphorylation-related genes. This trade-off between immunity and metabolism is consistent with findings in other insects and vertebrates, where immune activation often leads to mitochondrial remodeling or dysfunction, either as a host-driven adaptation or as a result of pathogen interference[37–39].

In our study, trophocytes transitioned from a basal state (cluster T1) to a highly active state (cluster T5) characterized by dramatic changes in gene expression (Fig. 6A–E, Supplementary data 7, 8). Notably, this shift was reversible by six days post-feeding, suggesting a temporary transcriptional reprogramming rather than permanent differentiation. Endoreplication is a modified cell cycle where there is DNA replication without cell division, leading to cellular polyploidy. Trophocyte endoreplication is well-documented in mosquitoes and other insects, serving to increase transcriptional output in support of reproduction[12,40–42]. Our findings reinforce this model by providing both molecular and histological evidence, as we observed EdU-positive, but pH3-negative trophocytes (Fig. 6G, H), consistent with DNA replication in the absence of mitosis. This aligns with a pronounced transcriptional induction of genes involved in DNA replication and cell cycle regulation at 24 h post-blood feeding (Supplementary Fig. 8, Supplementary data 8). These data suggest that endoreplication is a mechanism that enables a rapid response to the extreme metabolic demands that allows trophocytes to amplify expression of yolk protein genes and other genes involved in oogenesis. More broadly, these results highlight the dynamic and reversible transcriptional plasticity of the fat body and emphasize the importance of considering endocycling states when interpreting insect single-nucleus transcriptomic data.

Ultrastructural studies of the fat body in *Aedes aegypti* reveal significant changes in trophocytes as they transition from the pre-vitellogenic to the vitellogenic phase following blood feeding. During this transition, ribosomes that were initially free in the cytoplasm associate with the endoplasmic reticulum (ER), the Golgi complex enlarges and associates with the rough endoplasmic reticulum (RER), and the surface of the plasma membrane increases by forming more invaginations[7,12]. Interestingly, these changes are more pronounced in trophocytes adjacent to the hemolymph, suggesting that these cells are more actively engaged in protein synthesis and secretion towards the hemolymph[7,12], and agrees with the localized expression of Vg mRNA we observed in this trophocytes by ISH.

Furthermore, at a subcellular level, *Vg* transcripts are localized in the apical region of the trophocyte layer facing the hemolymph. In mammals, mRNA localization is often regulated by RNA-binding proteins associated with the cytoskeleton, influencing translation efficiency through ribosome positioning[43,44]. Similar regulatory mechanisms may be at play in trophocytes, enabling efficient protein export into the hemolymph. Notably, higher expression of *Vg* is observed in the fat body associated with the lateral body wall. This spatial pattern may be influenced by the substantial mechanical stretching that the mosquito lateral body wall undergoes during blood feeding, potentially affecting local gene expression. The lateral lobes of the fat body are also more prominent[45], which may increase surface area contact with the hemolymph, thereby enhancing Vg export.

In summary, our snRNA-seq analysis reveals the remarkable transcriptional complexity and functional specialization of the mosquito fat body in response to blood feeding and infection. By overcoming technical barriers to nucleus isolation, we uncovered a dynamic interplay between metabolic and immune pathways across multiple fat body-associated cell types. Hemocytes appear to be locally enriched and transcriptionally responsive, suggesting a model of tissue-resident immune surveillance. Trophocytes exhibit distinct transcriptional states, with a reversible transition toward reproductive specialization and endoreplication in response to blood feeding. Notably, the polarized expression of vitellogenin mRNA and metabolic shifts underscore the spatial and functional plasticity of fat body trophocytes. Together, these findings support a model in which the fat body integrates metabolic, reproductive, and immune functions through spatially organized and temporally regulated transcriptional programs. This study provides a high-resolution atlas of the mosquito fat body and associated cells, revealing key physiological responses to blood feeding, bacterial challenge, and immune priming, which establishes a foundation for future efforts to manipulate vector competence by targeting tissue-specific or state-dependent gene expression.

## Method

### Ethics statement
Public Health Service Animal Welfare Assurance #A4149-01 guidelines were followed according to the National Institutes of Health Animal (NIH) Office of Animal Care and Use (OACU). These studies were done according to the NIH animal study protocol (ASP) approved by the NIH Animal Care and User Committee (ACUC), with approval ID ASP-LMVR5.

### Mosquito rearing
*Anopheles gambiae* mosquitoes (G3 strain – CDC) were maintained under standard insectary conditions at 28 °C, 80% relative humidity, and a 12:12 h light:dark cycle. Adult stages were fed *ad libitum* with cotton pads soaked in 10% Karo syrup. Eggs were hatched in plastic trays containing 1 L of deionized water supplemented with a pinch of powdered koi fish food and 10 mL of yeast solution (prepared by dissolving two tads of live baker's yeast in 50 mL of deionized water). Larvae stages were reared in trays with 1 L of deionized water and fed every other day with fish food pellets (Tetramin, Blacksburg, VA, USA). The colony was maintained by feeding adult female mosquitoes on defibrinated cow blood using an artificial membrane feeding system.

### Blood feeding
Female *An. gambiae* mosquitoes, 4-5 days post-eclosion, were blood-fed on 3- to 4-week-old female BALB/c uninfected mice (Charles River, Wilmington, MA, USA) for 15 min. Prior to feeding, mosquitoes were starved for ~16 h. Only fully engorged females were selected and maintained on 10% karo syrup following feeding until sample harvesting. Same-age female mosquitoes maintained on *ad libitum* 10% karo syrup were used as a sugar-fed control. Abdominal fat body tissues were dissected 24 h post-blood meal, along with their age-matched sugar-fed control mosquitoes. Mice were housed in individually ventilated cages at 20–21 °C and 56% relative humidity with a 12:12 hour light-dark cycle and provided with sterile water and food *ad libitum*.

### *Plasmodium berghei* infection
Mosquito infections with *Plasmodium berghei* were carried out using the transgenic GFP-expressing ANKA GFPcon 259cl2 parasite line[46], maintained by serial passages into 3- to 4-week-old female BALB/c mice from frozen stocks. Prior to mosquito feeding, parasitemia was monitored using in vitro exflagellation assays as previously described[47]. Four-to-five-day-old female mosquitoes were allowed to feed on infected mice once parasitemia reached 3–6% and 1–2 exflagellations per field were observed. Same-age uninfected mice were used to feed control group of blood-fed mosquitoes. After feeding, both control and infected mosquitoes were maintained at 19 °C, 80% humidity, and 12:12 hour light:dark cycle. After 48 h, mosquitoes were shifted to 28 °C, under the same humidity and light conditions, to reduce oocyst development. Female mosquitoes were allowed to lay eggs at 4 days post-blood meal. Abdominal fat body tissues from control and infected samples were dissected at 6 days post-blood meal. A subset of 10–20 mosquitoes from each experiment was used to confirm the infection success by either: (1) oocyst counts at 10 days post-infection, as a direct measurement of infection; or (2) serpin 6 expression at 24 h post-infection as marker associated with midgut invasion by *P. berghei* ookinetes.

### Bacteria challenge
*Escherichia coli* and *Micrococcus luteus* were kept as frozen stocks at −80 °C. A pre-inoculum for each bacterium was set up and allowed to grow overnight at 37 °C, 250 rpm in a shaker incubator. On the day of the experiment, the pre-inoculum was diluted in fresh LB media and allowed to grow in the same conditions until $OD_{600}$ of 0.5. *E. coli* and *M. luteus* cultures were mixed at a 1:1 ratio and washed with sterile PBS to remove toxins. Adult 4-day-old female mosquitoes were injected with 138 nL of sterile PBS or bacteria mix. Abdominal fat body tissues from control and infected samples were dissected at 4 h post-injection. Validation of immune activation of the fat body after bacteria injection was confirmed by qPCR of antimicrobial peptides at 4 h post-injection.

### Single nuclei isolation
Female mosquitoes were cold-anesthetized on ice and placed on a drop of filter-sterilized adult hemolymph-like saline (HLS) buffer (Calcium chloride 2 mM, Potassium chloride 5 mM, HEPES pH7.4 5 mM, Magnesium chloride 8.2 mM, Sodium chloride 108 mM, Sodium bicarbonate 4 mM, Sodium phosphate 1 mM, sucrose 10 mM, Trehalose 5 mM −[48]). To isolate the abdominal fat body tissue, the abdomens were separated from the thorax. The gut, ovaries, and Malpighian tubules were then removed from the body cavity, leaving the abdominal fat body attached to the cuticle. For each experimental condition, two pools of 25 tissues were dissected and gently squeezed on ice in HLS buffer with a plastic pestle. The samples were washed twice with 80% methanol to fix and remove lipids. Centrifugations were carried out for 5 min at 500 × *g* at 4 °C. To release the nuclei, 500 μL of lysis buffer (1 mL Nuclei PURE lysis buffer, 1ul 1 M DTT, and 10uL Triton X-100; Sigma, MO, USA) was added to each sample and homogenized using a pre-cooled 1 mL Dounce tissue grinder, applying 15 strokes with both the loose and tight pestle. Samples were gently mixed with 1.8 mL of a 1.8 M sucrose cushion solution (5.4 mL Nuclei PURE 2 M Sucrose Cushion solution and 0.6 mL Nuclei PURE Sucrose Cushion Solution; Sigma, MO, USA) to dilute the lysis buffer. The sample was then subjected to a one-step continuous sucrose cushion for further purification. Each sample was divided into two tubes, with 900 μL of the homogenized sample carefully layered over 500 μL of

1.8 M sucrose cushion solution. Blood-fed samples were divided into four tubes. Centrifugation was carried out at 13,000 × g for 45 min at 4 °C. The supernatant was gently discarded, and the resulting pellet, containing the nuclei, was washed twice with washing buffer (1% BSA in PBS + 0.2 u Protector RNase Inhibitor; Sigma, MO, USA). All solutions were kept on ice during the procedure. Finally, the nuclear suspension was stained with 0.4% trypan blue, and the nuclear yield was estimated using a hemocytometer. Nuclei quality was accessed through light microscopy by the evaluation of nuclei membrane integrity. The targeted concentration of nuclear suspension was 700–1200 nuclei per microliter. Two biological replicates of the naïve and *P. berghei* mediated priming samples were collected. One biological replicate was performed for the control PBS-injected, bacteria-injected, sugar-fed, and blood-fed samples. For the detailed protocol please refer to de Carvalho et al.[49].

### Chromium 10X snRNA-seq library preparation and sequencing
After performing the single nuclei isolation of the fat body of *An. gambiae* female mosquitoes with different stimuli, an estimated number of 10,000 fresh single nuclei were immediately used for 10X Genomics Chromium droplet snRNA-seq. Single nuclei encapsulation, cDNA synthesis, and libraries preparation were carried out using the Chromium Next GEM Single Cell 3′ Reagent Kits v3.1 (10x Genomics, Pleasanton, CA), following the manufacturer's instructions. Assessment of cDNA and library quantification and quality was performed using 4200 Agilent TapeStation System with a High Sensitivity D5000 ScreenTape (Agilent, Santa Clara, CA). Paired-end sequencing was performed with Illumina NextSeq 550 or NextSeq 2000 platforms.

### Single-nucleus RNA-seq processing and ambient RNA correction
To generate the single-nuclei gene expression matrix, raw sequencing reads were aligned to the *Anopheles gambiae* PEST (AgamP4.14, v59) reference genome using the Cell Ranger pipeline (version 7.0.0, 10× Genomics, Pleasanton, CA). The reference genome and corresponding annotation file were obtained from VectorBase (https://vectorbase.org/common/downloads/Current_Release/AgambiaePEST). The gene annotation (gtf) file was filtered to retain protein-coding genes, and a custom reference was built using the "cellranger mkref" function. Alignment and quantification were performed with the "cellranger count" function, using the parameters "--include-introns=true" and "--expect-cells=10000", to generate both raw and filtered gene-barcode matrices. As nuclei isolation can result in the presence of ambient RNA—potentially from lysed cells—that becomes incorporated into droplets containing intact nuclei, we applied ambient RNA correction to the dataset. Correction was performed in each sample using the decontX algorithm from the celda package (version 1.20.0)[50], which models and removes contamination from ambient background RNA in snRNA-seq data.

### Quality control and cell filtering
Quality control and cell filtering were performed using the Seurat R package (version 4.2.1). For each sample, nuclei were retained based on sample-specific thresholds for the number of detected genes (Features), the percentage of mitochondrial gene expression, and the percentage of ribosomal gene expression, to remove low-quality nuclei and potential doublets. Cells were included if they expressed between 200 and 2000 genes for most samples (Naïve 1, Naïve 2, Prime 1, Prime 2, and Blood-fed), with adjusted upper bounds for PBS-control, bacteria-injected (2500), and Sugar-fed samples (1500). To minimize the inclusion of damaged or stressed nuclei, cells with a mitochondria gene percentage exceeding 2.5% (Naïve 1 and Prime 1), 1.5% (Naïve 2 and Prime 2), 7% (PBS-control), 5% (Bacteria-injected), or 10% (Sugar-fed and Blood-fed) were excluded. Additionally, a ribosomal gene expression threshold was applied, with upper limits ranging from 10% to 25% across samples to account for potential variation in

ribosomal content between conditions. Cellular complexity, assessed using the log10GenesPerUMI metric, was also evaluated to ensure transcriptomic diversity and minimize amplification bias; all retained nuclei had a complexity score greater than 0.75. These filtering parameters were chosen to optimize the retention of high-quality nuclei while minimizing technical artifacts and ambient RNA contamination.

### Integrative analysis, cell clustering, and marker identification
Seurat-based integration and clustering analysis was performed on previously described datasets: Naïve and Prime, PBS-control and bacteria-injected, and Sugar-fed and Blood-fed. This was achieved by first log-normalizing the gene counts of the remaining filtered nucleus using the "NormalizeData" function. Next, the top 2000 most variable genes were identified for each sample using the "FindVariableFeatures" function and tested for suitability as integration anchors using the "FindIntegrationAnchors" function. A final set of 2000 anchors were then used in the "IntegrateData" function to integrate the samples into a single dataset. Following integration, the dataset was scaled using the "ScaleData" function. Dimensionality reduction was performed using "RunPCA", and the first 30 principal components were used to generate a two-dimensional embedding with "RunUMAP". Clustering was conducted with "FindNeighbors" and "FindClusters", applying a resolution 0.3 to identify transcriptionally distinct cell populations. Marker genes for each cluster were identified using "FindAllMarkers" function in Seurat, requiring that genes must be detected in at least 25% of cells of the integrated dataset, an average log2 fold change (Avg log2FC) ≥ 0.5 and, and an adjusted *p*-value (P val adj) <0.05. Differential gene expression analysis was performed using the "FindMarkers" function to assess transcriptomic changes between treatment groups. For comparisons between PBS-control and bacteria-injected, as well as Sugar-fed and Blood-fed samples, stringent thresholds were applied, an average log2 fold change (Avg log2FC) ≥1 or ≤ −1 and a *p* adjusted value (P val adj) <0.001. A more permissive threshold, Avg log2FC ≥ 0.5 and P val adj <0.05, was used for comparisons between *P. berghei*-infected and naïve samples. Raw sequencing data have been deposited in the SRA database under BioProject accession number PRJNA1288431. The estimated relative cell abundance provided in Supplementary data 2 represents the relative proportion (%) of nuclei from each cluster relative to the total nuclei captured in each sample.

### RNA velocity analysis
RNA velocity analysis was performed by first generating spliced and unspliced transcript counts from 10x Cell Ranger outputs using velocyto v0.17.15[51]. Velocities were then computed and projected onto the Seurat-derived UMAP embeddings using scVelo v0.2.5[52] and its dynamical model. Data were preprocessed, filtered, and normalized with "pp.filter_and_normalize" (min_shared_counts=30, n_top_genes=2000), followed by computation of first and second moments using "pp.moments" (n_pcs=30, n_neighbors=30). Finally, "scv.tl.recover_dynamics", "scv.tl.velocity", "scv.tl.velocity_graph", and "scv.tl.recover_latent_time" were used to model transcriptional dynamics, estimate RNA velocities, construct directed cell-state transition graphs, and order cells along a latent developmental timeline.

### Bulk RNA-seq RNA extraction, library preparation, and sequencing
Body wall tissues (abdominal fat body attached to the cuticle) were dissected from adult female mosquitoes as previously described. Each biological replicate consisted of a pool of 10 tissues (*n* = 3 or 4). Dissected tissues were immediately transferred to 300 μL of TRIzol™ reagent (ThermoFisher Scientific, MA, USA) and stored at −80 °C until processing. Tissues were homogenized using an electric homogenizer, followed by the addition of 700 μL of TRIzol™ reagent. Samples were centrifuged at 12,000 × g for 5 min at 4 °C to pellet the cuticle. After a 5-min incubation at room temperature, 200 μL of chloroform was

added, and samples were vortexed briefly. Following a 3-min room temperature incubation, samples were centrifuged at $12,000 \times g$ for 15 min at 4 °C. The aqueous phase was transferred to a fresh 1.5 mL microcentrifuge tube, and the RNA was precipitated by adding 500 μL of isopropanol. Samples were gently mixed by inversion, incubated on ice for 15 min, and centrifuged at $12,000 \times g$ for 15 min at 4 °C. The resulting pellet was washed with 1 mL of 75% ice-cold ethanol, briefly vortexed, and centrifuged at $7500 \times g$ for 5 min at 4 °C. Supernatant was discarded, and pellets were air-dried before resuspension in 30 μL of nuclease-free water. RNA was homogenized and incubated at 55 °C for 5 min to enhance solubilization. Final RNA samples were stored at −80 °C. RNA libraries were prepared with NEB Next Ultra II RNA Library prep kit (New England Biolabs), and sequencing was performed on Illumina NovaSeq 6000 sequencer (150 bp paired-end reads) by Novogene (Sacramento, CA). Raw sequencing data have been deposited in the SRA database under BioProject accession number PRJNA1288431.

### Bulk RNA-seq data and statistical analysis

The quality of raw Illumina reads was checked using the FastQC tool (version 0.12.0) (https://www.bioinformatics.babraham.ac.uk/projects/fastqc/). Low-quality sequences with a Phred quality score (Q) below 20 and the Illumina adaptors were removed using TrimGalore v.0.6.7 (https://github.com/FelixKrueger/TrimGalore). High-quality reads were aligned to the *Anopheles gambiea* PEST (AgamP4.14, v59) transcriptome using RSEM v.1.3.3 (Li and Dewey 2011) with Bowtie 2 as the aligner for transcript quantification. The reference transcriptome was obtained from VectorBase (https://vectorbase.org/common/downloads/Current_Release/AgambiaePEST). Downstream analysis, including multi-dimensional scaling (MDS) plots and pairwise differential expression analysis were carried out with the edgeR package for R (version 3.19). For the differential expression analysis, the raw read counts of each transcript were used as input for edgeR, followed the standard TMM-normalization. Differentially expressed transcripts were identified using a negative binomial model, with log-fold-change (logFC) serving as the effect size and false discovery rate (FDR) controlling for multiple testing. Transcripts were considered differentially expressed if $|logFC| \geq 1$ and FDR < 0.05. For comparisons between naïve and *P. berghei*-challenged samples, a relaxed threshold of logFC ≥ 0.5 with FDR < 0.05 was applied.

### In situ hybridization (ISH)

Female mosquitoes were cold-anesthetized and placed on a drop of PBS. The abdomens were detached from the thorax and the midgut, ovaries, and Malpighian tubules were carefully removed from the body cavity. The body wall, including the abdominal fat body attached to the cuticle, was then incised along the lateral side using fine scissors. The dissected tissues were fixed for 1 h in 4% paraformaldehyde on a 9-well plate at room temperature. After fixation, the tissues were gently transferred to a 1.5 mL microcentrifuge tube and subjected to a series of washes. Tissues were washed three times for 5 min each with 1 mL of PBST (0.1% Triton X-100 in PBS), with gentle rocking at room temperature. To reduce lipid content, the tissues were washed with increasing concentrations of methanol (PBST: Methanol; 7:3, 1:1, 3:7) for 5 min at room temperature. An absolute methanol wash was then performed for 10 min, after which the tissues were rehydrated by washing with decreasing concentrations of methanol (PBST: Methanol; 3:7, 1:1, 7:3) for 5 min each. Finally, tissues were washed four times with PBST for 5 min each. ISH was performed using the RNAscope Multiplex Fluorescent Reagent Kit v2 assay (ACDBio, Abingdon, United Kingdom), adapted from the protocol described by [21]. Tissues were permeabilized with Protease Plus, provided by the kit, for 30 min at room temperature. Following two washes with PBST for 5 min, hybridization was carried out for 2 h at 40 °C. Amplification of the hybridized probes signal step was performed in sequential steps: AMP1 and AMP2 for 30 min each and AMP3 for 15 min, all at 40 °C. Between each amplification step, tissues were washed twice with wash buffer for 2 min at room temperature. Fluorescent signal development was achieved using the TSA-based Opal 4-Color Automation IHC kit (PerkinElmer, MA, USA). RNA probes specific to the target genes were synthesized by ACDBio (Abingdon, United Kingdom). Fluorescent labeling was caried out stepwise, with the addition of corresponding HRP (C1, C2, C3, or C4) for 15 min at 40 °C. Following each HRP incubation, tissues were washed twice for 2 min before incubation with the Opal fluorophore for 30 min at 40 °C. After each fluorophore incubation, tissues were washed twice for 2 min, and HRP activity was blocked by a 15-min incubation with HRP blocker at 40 °C. For morphological visualization, tissues were stained with Hoechst (1:10,000) to label the nuclei, and BODIPY 488 nm (1:5000 – ThermoFisher Scientific, MA, USA) to visualize neutral lipids in wash buffer for 10 minutes. Following a final wash to remove excess Hoechst and BODIPY, slides were mounted in a drop of ProLong™ Gold Antifade Mountant (ThermoFisher Scientific, MA, USA). Tissues were gently positioned and flattened in Prolong under a dissecting microscope to avoid folding or overlapping, and slides were allowed to dry overnight at room temperature in the dark.

### 5-ethynyl-2'-deoxyuridine (EdU) assay and Immunofluorescence

Female *An. gambiae* mosquitoes, 4–5 days post-eclosion, were blood-fed on 3- to 4-week-old female BALB/c uninfected mice (Charles River, Wilmington, MA, USA) for 15 min. Prior to feeding, mosquitoes were starved for ~16 h. 5-ethynyl-2'-deoxyuridine (EdU – ThermoFisher Scientific, MA, USA) 3 mg/mL 69 nL injections were performed at 6 h post-feeding. Only fully engorged females were selected and maintained on 10% karo syrup following feeding until sample harvesting. Same-age female mosquitoes were also injected with 69 nL of EdU 3 mg/mL and maintained on *ad libitum* 10% karo syrup were used as a sugar-fed control. Abdominal fat body tissues were dissected 24 h post-blood meal or 18 h post EdU injection, along with their age-matched sugar-fed control mosquitoes. Body walls were dissected in PBS and immediately placed in 4% paraformaldehyde for 30 min at room temperature, briefly transferred to ice-cold 80% ethanol for 3 min, and fixed again in 4% paraformaldehyde for 1 h at room temperature. Tissues were washed three times with PBST 0.1% (1× PBS, 0.1% Triton X-100) for 10 min each. EdU was developed as per manufacture instruction with the Click-iT Plus Plus EdU Imaging kit (ThermoFisher Scientific, MA, USA). Briefly, the Click-iT Plus reaction (1× Click-iT reaction buffer, Copper protectant, Alexa Fluor picolyl azide 594 nm, 1× Click-iT EdU buffer additive) was prepared and incubated with the sample for 30 min at room temperature. Tissues were washed three times with PBST 0.1% for 5 min. After this, tissues were blocked for 2 h at room temperature with blocking buffer (1× PBS, 0.1% Triton X-100, 2% BSA, 0.1% gelatin). Primary antibodies were diluted in blocking buffer: anti-phospho-histone H3 (pH3) (Ser10) (Millipore Sigma, MA, USA) 1:1000 or anti-Leucin rich protein 8 (LRR8) (Pacific Immunology, CA, USA) 1:100 and incubated overnight at 4 °C. The next day, body walls were washed three times with blocking buffer for 10 min and then incubated with goat anti-rabbit immunoglobulin G (IgG)–Alexa Fluor 555 nm (Invitrogen, CA, USA) (1:1000) diluted in blocking buffer at room temperature for 2 h. Tissues were washed twice in blocking buffer and then twice more in PBST 0.1%. Finally, tissues were incubated with Hoechst 405 nm 1:10,000 and BODIPY 488 nm 1:5000 and Phalloidin 647 nm (1:40) before being placed in ProLong Gold mounting media (Invitrogen, CA, USA).

### Confocal microscopy and image processing

Confocal images were captured using a Leica TCS SP8 (DM8000) confocal microscope (Leica Microsystems, Wetzlar, Germany) with either a 20× or a 40× oil immersion objective equipped with a

photomultiplier tube/hybrid detector. Leica Application Suite X (LAS X) software was used to capture microscopy images. Abdominal fat bodies were visualized with a white light laser, using 488 nm excitation and 52 nm emission for BODIPY 488 dye (neutral lipids), 550 nm excitation and 570 nm emission for Opal 570 (ISH probes), 588 nm excitation and 620 nm emission for Opal 620 (ISH probes), 676 nm excitation and 690 nm emission for Opal 690 (ISH probes), and a 405 nm diode laser for nuclei staining (Hoechst 33342). Images were taken using sequential mode and variable z-steps. Image processing and merging were performed using Imaris version 10.0.0 (Bitplane, MA, USA) and Adobe Photoshop CC (Adobe Systems, CA, USA).

## FAS mRNA quantification by in situ hybridization

To quantify the *FAS* mRNA fluorescence, we used the surface mode on Imaris version 10.0.0 (Bitplane, Concord, MA,USA). Using this method, we can calculate the fluorescence intensity per voxel for each object that is positive for the *FAS* staining (Source Data). As an input we used confocal images that were captures using the same laser power and gain between naïve and challenged tissues. To create a surface, we used confocal z-stack sections and first applied a threshold between 13 and 45 for the number of voxels. After the surfaces were generated, we applied a final filtering step removing volumes that were below 1024 $\mu m^3$ to remove smaller volumes and other non-related surfaces that might be generated. The fluorescence intensity mean per voxel was calculated for every surface that was positive for the mRNA of *FAS*. Statistical analysis was performed with GraphPad prism (version 10) using unpaired Mann−Whitney two-tailed test from 6 to 7 fields per condition ($n = 106$ oenocyte clusters in naïve samples; $n = 105$ oenocyte clusters in challenge samples).

## Statistical analysis and reproducibility

No statistical methods were used to predetermine sample size. Sample sizes were chosen based on prior literature and practical considerations. No data was excluded except during standard quality control for sequencing analyses. Experiments were not randomized, and investigators were not blinded during experiments or outcome assessments. Single-nucleus RNA-seq analyses were performed in Seurat using predefined thresholds for gene detection, effect size, and adjusted $P$ values, with multiple testing controlled by the Benjamini−Hochberg method. Differential expression analyses used nonparametric tests implemented in Seurat, with comparison-specific significance thresholds as described in the Methods. RNA velocity analyses were conducted using velocyto and scVelo with the dynamical model, without additional hypothesis testing. Bulk RNA-seq differential expression was analyzed using edgeR following TMM normalization, modeling counts with a negative binomial distribution and applying false discovery rate correction. Fluorescence intensity from in situ hybridization experiments was quantified using Imaris, and statistical comparisons were performed using unpaired two-tailed Mann−Whitney tests in GraphPad Prism 10. Exact sample sizes, statistical tests, and significance thresholds are provided in the figure legends and Supplementary Tables. All experiments were conducted using consistent parameters to ensure reproducibility.

## Reporting summary

Further information on research design is available in the Nature Portfolio Reporting Summary linked to this article.

## Data availability

Raw fastq files from single nucleus and bulk RNA sequencing are available the SRA database under the Bioproject number PRJNA1288431. All other relevant data are provided within the paper and its Supplementary files. Source data are provided with this paper.

## Code availability

The code supporting the findings of this study has been deposited in Zenodo and is accessible via the DOI: 10.5281/zenodo.17886096 [https://zenodo.org/records/17886096]. The repository contains all analysis scripts and documentation required to reproduce the results.

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

## Acknowledgements

We thank Kevin Lee, Yonas Gebremicale, and André Laughinghouse for insectary support. We thank Nathania Dabila and Mu Jianbing for sequencing support. This research was supported by the Intramural Research Program (Z01AI000947 – C.B.-M.) of the National Institutes of Health (NIH). The contributions of the NIH author(s) were made as part of their official duties as NIH federal employees, are in compliance with agency policy requirements, and are considered Works of the United States Government. However, the findings and conclusions presented in this paper are those of the author(s) and do not necessarily reflect the views of the NIH or the U.S. Department of Health and Human Services.

## Author contributions

Conceptualization: S.S.D.C., C.B.-M., Methodology: S.S.D.C., A.B.F.B., C.M., Investigation: S.S.D.C., A.B.F.B., C.M., Visualization: S.S.D.C., A.B.F.B., Funding acquisition: C.B.-M., Project administration: C.B.-M., Supervision: C.B.-M., Writing—original draft: S.S.D.C., C.B.-M., Writing—review & editing: S.S.D.C., C.B.-M.

## Funding

## Competing interests

The authors declare no competing interests.
