## [Transparent Peer Review file · Nature Communications]

Cell-Specific Responses of *Anopheles gambiae* Fat Body to Blood Feeding and Infection at Single-Nuclei Resolution

Corresponding Author: Dr Carolina Barillas-Mury

Version 0:

Reviewer comments:

Reviewer #1

(Remarks to the Author)
General review

This manuscript presents a comprehensive single-nucleus RNA-seq atlas of *Anopheles gambiae* abdominal body wall-associated tissues under various physiological and immune challenge conditions, including blood feeding, bacterial injection, and *Plasmodium* infection. An innovative delipidation-based nuclei isolation protocol facilitated snRNA-seq analysis of this lipid-rich tissue. The authors effectively integrate snRNA-seq with extensive RNA FISH to characterize six major abdominal fat body cell types, with a particular focus on trophocyte subpopulations and oenocytes, verifying condition-specific responses. The study highlights the role of lipid-specialized oenocytes in immune priming during *Plasmodium* infection. Additionally, it demonstrates transcriptional responses of trophocyte subpopulations to bacterial challenges, accompanied by hemocyte expansion. Moreover, the research reveals that blood feeding induces hemocyte proliferation and a reversible trophocyte transition into a vitellogenic state. This study is an in-depth characterization of the cellular composition of mosquitoes fat body tissues and their role upon infection, making a significant contribution to our understanding of mosquito physiology and immunity. It employs state-of-the-art techniques for single-nuclei analysis and functional validation; the manuscript is well-written and data well presented.

Please find below some major and minor comments that will hopefully help the authors to further improve their work:

Major comments

1- The study aims to characterize *Anopheles gambiae* fat body responses to bacterial and *Plasmodium berghei* infections. However, the authors do not provide evidence of successful infections or the prevalence of infection in the mosquitoes used for snRNA-seq, bulk RNA-seq, and RNA FISH. Including data on infection rates, such as oocyst/sporozoite counts or parasite qPCR for *Plasmodium*, and CFU or qPCR for bacteria at sampled time points, would clarify the prevalence of infection in each condition and strengthen the study.

2- For bacterial infections, authors might provide justifications as to (1) Why bacteria were introduced via injection rather than through a natural exposure, which might better mimic natural infection processes and (2) why a mixture of 2 bacterial strains was employed over a single strain infection. Clarifying where the bacteria were injected and discussing any potential differences in immune response compared to natural gut passage would provide valuable context. Determining if bacteria can be detected within the fat body itself would also enhance the understanding of localized infection dynamics.

3- Have the authors considered the possibility of detecting pathogen-derived transcripts, in addition to host transcript, in snRNA-seq and bulk RNA-seq datasets? Furthermore, RNA FISH could be used for parasite or bacteria detection to confirm the presence of pathogens and validate infection status in the study.

4- In reference to the statement about FAS expression being "highly induced in response to priming in oenocytes," the authors should consider providing quantification of their ISH measurements. Including statistical analyses would strengthen claims by offering more than visual intensity differences, particularly for validating changes seen in Figure 3.

5- The authors specify the concentration of the nuclei suspension used for sn-RNAseq, but some additional key information would be interesting to share : (1) what is the yield of the nuclei isolation, i.e. the number of nuclei retrieved compared to the total number of expected nuclei ? (2) How did the authors verify nuclei integrity and quality ?

6- In Figures 6G-H, authors write « A few hemocytes were also EdU+ in both sugar-fed and blood-fed mosquitoes ». Could the authors please explain how they can identify those cells as hemocytes in the absence of specific staining ?

Minor comments

7- Given the context that "transcriptional differences between trophocyte clusters are more subtle and unique markers cannot be used to identify specific subpopulations," the authors might consider using trajectory or velocity analysis, to explore whether these clusters represent specific subtypes or continuous states.

8- The authors should consider employing differential expression analysis by using pseudo-bulk methods, such as DESeq2 or edgeR, on the single-nucleus RNA-seq data. This approach could help identify the same upregulated genes observed in bulk RNA profiles under conditions of Plasmodium infection, bacterial challenge, or blood feeding.

9- In Figure 1, panel A illustrates an overview of the experimental workflow, including both snRNA-seq and bulk RNA-seq. However, the rest of Figure 1 only presents results from snRNA-seq. It would be beneficial to include at least a general result from the bulk RNA-seq to align with what is indicated in panel A. In the legend for panel C, the statement "3 representative marker genes per cluster" is inaccurate for T2, which only includes 2 markers.

10- While the authors indicate that Galactokinase (GALK) serves as a marker for all trophocyte clusters (line 122-124), it does not appear highly expressed in T5 (Figure 1C, Figure 2B). This observation weakens the claim that GALK is a universal marker across all trophocyte profiles, regardless of blood feeding status. This, this claim could be moderated.

11- Fatty acid synthase (FAS) has multiple isoforms in some mosquito species. Could the authors clarify the specific FAS gene or isoforms measured in Anopheles gambiae? Are certain FAS isoforms uniquely enriched in oenocytes (line 138-139), or are they distributed across various cell types, including trophocytes? Understanding the specialization of FAS in oenocytes would provide valuable insights.

12- The results of the bulk transcriptomic analysis after bacterial challenge (lines 208–222), including the regulation of antimicrobial, immune, and metabolic pathways, are mentioned only in a supplementary table. The authors might consider adding a main or supplementary figure to enhance the impact and accessibility of these findings. In addition, the strong transcriptional response to blood feeding, (lines 286-310) is currently presented only in supplementary tables, which diminishes their impact. To enhance clarity and strengthen these findings, the authors might consider adding them in a main or supplementary figure.

13- In Figure 2A, the authors might add another item in the legend to specify what the grey boxes behind the central and dorsal tissues represent.

14- Would the author be able to clarify which study or facts supports the following claim : "lipids interfere with cell dissociation and 10X single-cell partitioning and Gel Bead-in-Emulsion (GEM) formation » ?

15- The authors might want to include a definition of endoreplication in their discussion.

Reviewer #2

(Remarks to the Author)

Reviewer #3

(Remarks to the Author)

In this article the authors present a well-executed set of experiments at single-nucleus resolution to characterise the complexity of the mosquito fat body—an organ central to both female reproductive development and innate immunity. The work is relatively descriptive in nature, as is often the case with this approach, but the level of cellular and molecular complexity revealed is valuable and will provide an important reference point for the field. It is therefore likely to be frequently cited.

The introduction is exemplary in its clarity. However, both I and a more junior colleague who helped me review the manuscript felt that some important detail was missing from the description of the experimental set-up used to examine the effects of immune priming. This is a field in which the authors have made many of the seminal contributions—indeed, they

effectively “wrote the book” on the subject—so the rationale deserves to be laid out more fully. The hypotheses underpinning these experiments could be stated more explicitly, and fewer assumptions should be made about the reader’s familiarity with why particular time points, bacterial delivery methods, and bacteria–Plasmodium combinations were chosen.

A similar point applies to aspects of the discussion concerning transcript localisation and endoduplication. The interpretations themselves are reasonable, but the arguments would be clearer with more explicit rationale and fewer assumptions about prior knowledge. A little more empathy for the reader in these explanations would substantially strengthen the manuscript.

More detailed comments and queries are provided below:

In the results section there needs to be an explanation of why they looked at the conditions they did. It would be helpful to add a justification of why both single nucleus and bulk sequencing was used (i.e. what different results were expected from each one). Also on the experimental rationale, the manuscript would be easier to understand if the authors add an explanation of why systemic bacterial challenge by injection into the hemolymph is an important and relevant immune challenge to mosquito biology, including explaining the choice of bacteria used in the challenge (perhaps representatives of gram positive/ negative?)

Several sections of the results presented interesting but incomplete stories. Adding either further experiments or at least fuller interpretation of the observations with clearer recommendations for future work would greatly improve the clarity and interest of the manuscript to a wider audience. These included:

1. Does the trade-off between immunity and metabolism mentioned on p18 of the MS (lines 393-397) extend to measurable impacts on reproduction (as explored for *Drosophila* in Gupta et al 2022 DOI: 10.1186/s12915-022-01328-w)? Importantly, how does bacterial or Plasmodium challenge impact the DNA replication or expression of yolk protein genes in trophocytes?

2. Vitellogenin mRNA localisation. A possible parallel with past observations in *Aedes* trophocytes entering the vitellogenic phase is mentioned (lines 412-420), with ribosomes relocating and plasma membrane changes. Figure 6 shows a clear localized expression of Vg mRNA in the trophocytes at the hemolymph side using in situ hybridisation. However, given that the single cell results were based on single nuclei, one might expect that Vg mRNA would additionally be enriched in the nuclei of these cells compared to the other cell types, since transcripts outside of the nucleus should have been removed during nucleus preparation and their data processing steps to remove ambient mRNA. In figure 6 Vg mRNA cannot be seen in trophocyte nuclei. It should be explained why this is. Is this because the level in the nuclei is low compared to the strong cytoplasmic accumulation, or is it due to contamination of the sequenced nuclei with non-nuclear mRNA?

Furthermore, there was a general issue with all in-situ microscopy presented that there was no marker for the cell membrane or sub membrane cytoskeleton, making it impossible to interpret where the cell margins were. Adding a cell outlining marker would really aid interpretation of these images for the reader and make the cell specific markers much more convincing.

3. More explanation of how cells were counted is needed. A fuller explanation of the rationale for counting cell types based on single nucleus sequencing and the error of these measurements (where replicate experiments allow this to be estimated) should be added to the methods section. This is needed to validate the many interesting observations on the changes in the proportions and numbers of cell typed during different treatments. For example, the story regarding circulating versus static granulocytes. In the introduction, lines 61-67 mention that immune priming in *An. gambiae* is mediated by release of hemocyte differentiation factor when microbiota contacts the midgut epithelia, and that this triggers an increase in the proportion of circulating granulocytes. Theoretically a changing proportion between circulating and fat-body associated granulocytes could be achieved either by release of fat body granulocytes to the haemolymph (resulting in fewer fat body granulocytes but more circulating), or by cell division which would boost the circulating population leaving the fat-body associated population at the same size. These alternative hypotheses could have been investigated using their combined single nucleus followed by in situ hybridisation methods. In their investigation of the response of fat body associated granulocytes to *P. berghei* challenge there was a clear transcriptional change but crucially, they state that they did not observe a significant difference in the number of body wall-associated granulocytes (results lines 181-182). This leaves the reader wondering how they measured the number of granulocytes. Was this based on the relative proportion of nuclei with a granulocyte expression profile compared to other fat body cell types in the single nuclei sequencing? If this is the case, what is the quality control for this – how do we know that the expression and physiological changes associated with the *P. berghei* challenge (or any of the other treatments used in the manuscript) did not influence the recovery rate of nuclei from different cell types differently and bias the count? Can this observation be confirmed using HCR-FISH and counting fat body associated granulocytes based on cell-type specific markers in naïve vs challenged mosquitoes? Would the final interpretation of this result be that the boost to circulating granulocytes is not by release of fat body associated granulocytes? This issue of how cells of each type were counted and how confident we can be in the counts/ relative proportions going on single nucleus clustering alone is also relevant for the stated observations about trophocytes (lines 263-265), granulocytes following bacterial challenge (lines 356-360) and oenocytes (p369-371).

4. The story regarding endoreplication in trophocytes was very interesting but seemed to finish before reaching a firm conclusion. They observed EdU-positive but pH3 negative trophocytes which would be consistent with DNA replication without mitosis. Why did they not attempt to look at the ploidy in these cells? This could be done using for example FACS or even just measuring the brightness of nucleic acid stains such as Hoechst using microscopy. Similarly, given that they applied a filter to removed doublets from single nucleus data, can they be sure they did not accidentally filter out trophocytes with a higher ploidy – how did the various QC metrics differ between potential doublets and endoreplicated nuclei?

Furthermore, in this study they only investigated the response of the fat body to the first blood meal. If the endoreplication occurs after the first blood feeding to boost transcription, this raises questions as to what happens to ploidy after 6 days when the transcriptional profile returns to pre-blood fed patterns, particularly in their cluster “T1” – are all these extra DNA copies kept? Or perhaps could the “T1” cluster 6 days after blood feeding be new cells and not the same T5 endoreplicated cells returning to pre-blood fed transcriptional patterns? If the cells in T1 are the T5 cells and ploidy remains at this boosted level, does that mean that after subsequent bloodmeals the transcriptional shift would be different since it should no longer

require genes involved in DNA replication since the endoreplication has already happened?

5. On the report of the result that hemocytes are the second most abundant cell type (lines 124-125) "Surprisingly, hemocytes (Hm1, Hm2) are the second most abundant cell type, representing 7.4% of the cells (Supplementary Table 2)", it needs to be explained why this is surprising?

6. The question of whether cells are "resident" or "associated" with the fat body does raise the question of whether they could move in and out. This is particularly relevant to the topic of the parallels between insect fat body and the vertebrate immune system (e.g. the potential role of T4 trophocytes in immune surveillance discussed lines 391-393), a topic that is of strong interest to the general reader. They postulate in lines 356-357 "We observed a doubling of the number of fat body-associated granulocytes in response to bacterial challenge, suggesting either proliferation of resident cells and/or recruitment from circulation". Although directly testing the alternative hypotheses of cell movement vs cell proliferation might be beyond the scope of this study, this question should be better highlighted in the discussion with a recommendation as to what sort of experimental approach could definitively answer this question in future studies (for example tracking of cells over time).

Further discussion of how the *Anopheles* fat body is similar or different from other insects would have improved the discussion section. This seems part of the stated aim of addressing a knowledge gap in Anopheline mosquitoes about the cellular composition and physiology of the fat body. The discussion section alludes to studies of the fat body cellular composition in other organisms including *Drosophila melanogaster* and *Bombyx mori*, and yet there is no discussion of the similarities and differences between what they observed in *Anopheles gambiae* fat body single nucleus sequencing and these earlier works.

There are a few technical considerations. As stated by my colleague above, the manuscript appears to be generally technically sound, but with the rationale of experiments, some technical issues warranted a clearer explanation. In the discussion the authors claim to have overcome technical barriers to nucleus isolation (line 330, 432). They give no details of these technical barriers and don't show any quality control of the nuclei isolated prior to sequencing (for example microscopy to show nuclei morphology). They seem to have followed initially a protocol similar to what was used in Gupta and Lazzaro 2022 (<https://doi.org/10.1080/19336934.2021.1978776>) followed by using the Sigma "Nuclei Pure Prep" standard protocol. Were other options for nuclei isolation tried and worked less well? As a minimum, images of the isolated nuclei for each treatment and yield as expressed as amount of nuclei/number of tissues dissected should be shown. It could also be acknowledged that *Anopheles* oenocytes have been isolated in the past (Grigoraki et al 2020, 10.7554/eLife.58019), although this was with the help of a transgenic GFP expressing line whereas the method developed in the current manuscript are applicable to all genotypes.

The number of biological replicates used for the single cell work was very low. Supplementary figure 1G suggests 2* "naïve", 2* "challenge", and only one each of other conditions (although "control" might be the same as one of them?). This low level of replication is likely due to the high cost of undertaking the single cell work but means that any important observations such as on the relative numbers of cells in each cluster need to be confirmed by another method for statistical robustness.

On the analysis of the single nucleus sequencing data, lines 102-103 state that the "cutoffs for the number of counts and features, as well as the percentage of mitochondrial and ribosomal content, were set for each sample (Supplementary Figure 1)". However, although supplementary figure 1 shows the spread of the data it does not actually indicate what cutoffs were used or how the decision to set these cutoffs was made. This should be made clear.

The selection of cluster specific markers seemed to have a very low bar in terms of the fold change (Avg log₂FC > 0.5) – just 1.41 times more expression in the enriched clusters? The authors should explain how this cut-off was decided on. In supplementary table 1 where these markers are described, the proportion of cells with detectable expression was also rather low for some marker genes.

Lines 531-533 mention that "The gene annotation (gtf) file was filtered to retain protein-coding genes". Could valuable data on the expression of non-protein coding genes such as long noncoding RNAs have been discarded at this point and how would its inclusion have affected the clustering?

For the ambient RNA correction (lines 537-538 "Correction was performed using the decontX algorithm from the celda package [48]" it should be specified if the same model was applied across all samples or whether this was modelled individually in each sample.

Reviewer #4

(Remarks to the Author)

de Carvalho and coworkers present a single nuclei RNAseq data set of *Anopheles gambiae* fat body. They show a variety of cell types, and notably assign five trophocyte clusters. They show some cells responding to bacterial challenge and, not surprisingly see a strong upregulation of vitellogenesis-related genes after a blood meal.

The introduction is beautifully written, the experiments are well described. The results show the expected heterogeneity of cell types in this tissue.

This is a well-done study.

Minor points:

- a scheme of where the different trophocyte types are located within the abdomen would be helpful.
- could the tissue and subcellular locations of some mRNAs you see (especially Vg) be due to technical problems of the in situ hybridization protocol? Just asking questions...
- a longer discussion of how the results of [30] compare to these findings would be appropriate.

Version 1:

Reviewer comments:

Reviewer #1

(Remarks to the Author)

"The authors have addressed all of our concerns comprehensively. They have conducted further analyses, generated new figures and thoroughly revised the text. The manuscript is now much stronger, more rigorous and clearer than the original submission. The major concerns regarding infection validation and methodological aspects have been resolved with compelling new data and arguments.

Based on these revisions, the manuscript is ready for publication, and we congratulate authors for this great piece of work."

Reviewer #2

(Remarks to the Author)

Reviewer #3

(Remarks to the Author)

The authors received extensive comments. I am happy with the changes they have made and detailed. Where these have not been included I am happy with the rebuttal. I thank the authors for patiently engaging with the process. This will be an important reference work for the field.

Reviewer #4

(Remarks to the Author)

I feel this manuscript is ready for publication.

Reviewer #1 (Remarks to the Author)

General review

This manuscript presents a comprehensive single-nucleus RNA-seq atlas of *Anopheles gambiae* abdominal body wall-associated tissues under various physiological and immune challenge conditions, including blood feeding, bacterial injection, and *Plasmodium* infection. An innovative delipidation-based nuclei isolation protocol facilitated snRNA-seq analysis of this lipid-rich tissue. The authors effectively integrate snRNA-seq with extensive RNA FISH to characterize six major abdominal fat body cell types, with a particular focus on trophocyte subpopulations and oenocytes, verifying condition-specific responses. The study highlights the role of lipid-specialized oenocytes in immune priming during *Plasmodium* infection. Additionally, it demonstrates transcriptional responses of trophocyte subpopulations to bacterial challenges, accompanied by hemocyte expansion. Moreover, the research reveals that blood feeding induces hemocyte proliferation and a reversible trophocyte transition into a vitellogenic state. This study is an in-depth characterization of the cellular composition of mosquitoes fat body tissues and their role upon infection, making a significant contribution to our understanding of mosquito physiology and immunity. It employs state-of-the-art techniques for single-nuclei analysis and functional validation; the manuscript is well-written and data well presented.

Thank you for the positive feedback.

Please find below some major and minor comments that will hopefully help the authors to further improve their work:

Major comments

1- The study aims to characterize *Anopheles gambiae* fat body responses to bacterial and *Plasmodium berghei* infections. However, the authors do not provide evidence of successful infections or the prevalence of infection in the mosquitoes used for snRNA-seq, bulk RNA-seq, and RNA FISH. Including data on infection rates, such as oocyst/sporozyte counts or parasite qPCR for *Plasmodium*, and CFU or qPCR for bacteria at sampled time points, would clarify the prevalence of infection in each condition and strengthen the study.

Response:

Thank you for pointing out the need to include the data we used to validate the *Plasmodium berghei* and bacterial infections. We have standardized protocols in our laboratory to validate both *Plasmodium* and bacterial infections, and this information is now included in the revised manuscript. We appreciate the opportunity to clarify these important aspects of our methodology.

***P. berghei* infections:** we used 2–3-week-old mice exhibiting parasitemia levels between 3–6% and an exflagellation rate of 1–2 events per microscopic field. These parameters were chosen to ensure a consistent and robust infection in the mosquito midgut.

Mosquito immune priming experiments: Our study specifically focused on the priming response triggered by *Plasmodium* infection, with particular interest in the role of oenocytes in the fat body. Ookinete invasion, in the presence of the gut microbiota triggers a systemic and permanent state of enhanced immunity. The establishment of the immune priming occurs within the first 24 hours post blood feeding, when ookinetes invade the mosquito midgut. Fat body tissue samples were collected 6 days post-infection to capture the long-term effect of priming on fat body trophocytes and oenocytes.

To confirm successful *Plasmodium* infection, we employ one of the two following strategies:

1. **Direct oocyst counts:** A subset of 10–20 mosquitoes were kept alive, and midgut dissections were performed at 10 days post feeding to directly count the number of oocysts/midgut, thereby validating the success of the infection. This method was used for the snRNAseq Exp #1 and the Bulk-RNAseq experiments (see figure below).

The disadvantage of this strategy is that we need to collect the samples 6 days post-infection and start the processing the sample before we know if the infection was successful (counting for *P. berghei* needs to be done 10-12 days post infection). To solve this problem, we can also use a molecular marker of midgut invasion.

2. **Molecular marker:** Serpin 6 expression is highly induced in ookinete-invaded midgut cells 24 h post-feeding and expression of this gene is a well-established marker of mosquito midgut infection by *P. berghei* ookinetes. The expression levels of *Serpin 6* midgut mRNA 24 hours post feeding between control and infected midguts was compared as an indicator of ookinete invasion. This strategy was used in snRNAseq Exp #2, and we confirmed that in that experiments *Serpin 6* expression was more than 5,000-fold higher in midguts infected with *P. berghei*, than in control females fed on an uninfected mouse (see figure below).

For bacterial infections, we injected a mixture of a Gram-positive (*Micrococcus luteus*) and a Gram-negative (*Escherichia coli*) bacterium directly into the mosquito hemolymph to induce a systemic immune

response. The insect fat body is the primary site of synthesis of antimicrobial peptides and proteins circulating in the hemolymph. Our rationale was to use a broad immune stimulus to activate systemic immune responses and assess the corresponding activity in the mosquito fat body. Previous studies have demonstrated that the fat body mounts a strong immune response to systemic infections [1]. This includes the transcriptional activation of antimicrobial peptides (AMPs) genes, which are well-established markers of immune pathway activation.

Mosquitoes were systemically infected using a fine glass needle inserted into a soft area of the thorax. Injected bacterial are quickly cleared from circulation and there are no reports of bacteria directly infecting trophocytes or other fat body-associated cell types.

To confirm that the fat body immune response was activated following the bacterial challenge, we measured the mRNA expression of some AMP genes by qPCR at 4 hours post-injection. We confirmed a strong response with an increase in expression of defensin 1 (200-fold) and cecropin 1 (135-fold), two well-characterized AMPs, as well as lysozyme 1 (35-fold) confirming that the bacterial injection successfully stimulated the fat body's immune response (results below).

These validation strategies to confirm the infections have been included in the methods section of the revised manuscript: “A subset of 10 - 20 mosquitoes from each experiment was used to confirm the infection success by either: (1) oocyst counts at 10 days post-infection, as a direct measurement of infection; or (2) serpin 6 expression at 24h post-infection as marker associated with midgut invasion by *P. berghei* ookinetes.” was included under the topic “*Plasmodium berghei* infection” in the methods section. And the sentence: “Validation of immune activation of the fat body after bacteria injection was confirmed by qPCR of antimicrobial peptides at 4 hours post-injection.” was included under the topic “Bacteria challenge” in the methods section.

2- For bacterial infections, authors might provide justifications as to (1) Why bacteria were introduced via injection rather than through a natural exposure, which might better mimic natural infection processes and (2) why a mixture of 2 bacterial strains was employed over a single strain infection. Clarifying where the bacteria were injected and discussing any potential differences in immune response compared to natural gut passage would provide valuable context. Determining if bacteria

can be detected within the fat body itself would also enhance the understanding of localized infection dynamics.

Response:

These are relevant questions that are explained in more detail in the revised manuscript. We appreciate the opportunity to expand on the rationale of our experimental design.

- (1) Our aim was to induce a broad systemic immune response and assess its direct impact on the fat body tissue. We choose bacteria injection for several reasons:

Controlled dosage: Injecting bacteria allows us to deliver a standardized number of bacterial cells into each mosquito, reducing variability in the immune response across individuals.

Direct exposure: By introducing the bacteria directly into the hemolymph, we ensure that the observed transcriptional changes in the fat body are the result of systemic immune activation.

Tissue specificity: An oral infection would primarily localize bacteria to the midgut, which is not the focus of this study. We agree with the reviewer that the responses to oral infection are important, however, the midgut has its own immune defenses to ingested microbiota or pathogens, which could introduce confounding variables and indirect effects on the fat body.

We used a sublethal bacterial load to activate the fat body response and we believe that a systemic injection was the best way to address the question of whether all trophocytes respond the same to a systemic bacterial challenge or whether there is some level of specialization.

- (2) We used a combination of two bacterial strains—one Gram-positive (*Micrococcus luteus*) and one Gram-negative (*Escherichia coli*)—to ensure broad activation of the major immune pathways in mosquitoes. Our goal was to provide a broad and robust immune stimulus capable of activating the systemic immune response in the fat body, allowing us to assess which fat body-associated cell types respond.

As noted previously, bacteria were injected into the hemolymph through a soft spot in the thorax. Importantly, the thorax was not included in the dissections or in our downstream analyses—only the abdominal fat body and body-wall associated tissues were examined. Thus, any immune response observed in the fat body reflects the systemic exposure to bacteria or bacterial immune elicitors circulating in the hemolymph, rather than local effects from the injection site.

Because the fat body is bathed by the hemolymph, trophocytes and other tissues directly respond to bacteria or immunogenic molecules released by bacteria.

To our knowledge, there are no reports describing bacteria inside trophocytes or other fat body-associated cell types, except for phagocytic hemocytes, in mosquitoes. While some studies have shown bacteria in close proximity to pericardial cells in the periosteal region [2], these bacteria were not observed within the cells themselves. Given that we injected bacteria into the thorax and excluded thoracic tissues from analysis, it is unlikely that bacteria were present within the fat body tissue examined in our study. Nonetheless, it is well established in several insect species that immune activation in the fat body leads to antimicrobial peptide (AMP) expression and secretion, which supports the use of AMPs gene expression as a readout for activation of immune response in this tissue.

We have now clarified this point on the results section: “The fat body plays an important role in the systemic immune response by producing antimicrobial peptides and contributing to pathogen melanization. However, cell type-specific responses remain poorly understood. The transcriptional response of the abdominal body wall to bacterial infection was explored by challenging adult females with a systemic injection of a mixture of a Gram-negative (*E. coli*) and a Gram-positive bacteria (*M. luteus*), or a sterile PBS control, and harvesting the abdominal body wall 4 hours later. This approach ensured the direct delivery of a standardized bacterial load into the hemolymph, triggering a robust and physiologically relevant immune response. The combination of Gram-positive and Gram-negative bacteria provided a broad immune stimulus, ensuring the activation of the major immune signaling pathways.”

3- Have the authors considered the possibility of detecting pathogen-derived transcripts, in addition to host transcript, in snRNA-seq and bulk RNA-seq datasets? Furthermore, RNA FISH could be used for parasite or bacteria detection to confirm the presence of pathogens and validate infection status in the study.

Response:

Thank you for suggesting this experiment.

We did not consider detecting pathogen-derived transcripts using either bulk RNA-seq, snRNA-seq, or RNA FISH from fat body tissues, as neither *Plasmodium* nor bacteria are expected directly associated with the fat body tissue. Parasites remain within the midgut basal lamina and studies with labeled bacteria have shown that they are quickly filtered out by hemocytes associated with the periosteal region of the mosquito heart. In another ongoing project, we are analyzing the responses of midgut tissues, and we are also analyzing the mRNAs from the parasites at the ookinete and oocyst stages.

4- In reference to the statement about FAS expression being "highly induced in response to priming in oenocytes," the authors should consider providing quantification of their ISH measurements. Including statistical analyses would strengthen claims by offering more than visual intensity differences, particularly for validating changes seen in Figure 3.

Response:

Thank you for suggesting statistical analysis of the RNA FISH images. We have now quantified the mean fluorescence intensity per oenocyte voxel in naïve and primed RNA FISH images (6-7 fields each) and included these results as Supplementary figure 4. The manuscript has been revised to emphasize the localization of FAS in oenocytes as revealed by ISH, and the quantification of FAS fluorescent signal intensity has been incorporated into the results section, that now reads: “Furthermore, we confirmed by ISH that FAS expression induction in response to priming occurs specifically in oenocytes present in the lateral and ventral regions of the abdominal wall (Figure. 3C-D, Supplementary Figure 3C-F, Supplementary figure 4). The induction in expression of FAS observed in bulk RNA-seq was also confirmed by ISH, which showed approximately a two-fold increase of FAS following priming (Supplementary figure 4).”

We have included the “FAS mRNA quantification by *in situ* hybridization” topic on the methods: “To quantify the FAS mRNA fluorescence, we used the surface mode on Imaris 10.0.0 (Bitplane, Concord, MA, USA). Using this method, we can calculate the fluorescence intensity per voxel for each object that is positive for the FAS staining. As an input we used confocal images that were captured using the same laser power and gain between naïve and challenged tissues. To create a surface, we used confocal z-stack sections and first applied a threshold between 13 and 45 for the number of voxels. After the surfaces were generated, we applied a final filtering step removing volumes that were below 1024 μm^3 to remove smaller volumes and other non-related surfaces that might be generated. The fluorescence intensity mean per voxel was calculated for every surface that was positive for the mRNA of FAS.”

5- The authors specify the concentration of the nuclei suspension used for sn-RNAseq, but some additional key information would be interesting to share: (1) what is the yield of the nuclei isolation, i.e. the number of nuclei retrieved compared to the total number of expected nuclei? (2) How did the authors verify nuclei integrity and quality?

Response:

The reviewer brings up an important point. To our knowledge, there is currently no published estimate of the total number of cells or nuclei in the mosquito abdominal fat body, which limits the ability to define an expected recovery yield. We worked for several months to optimize the recovery and quality of the nuclei. In our experience, sucrose cushion cleanup is the step with the highest loss of nuclei, but it is critical to remove debris and enrich for nuclei. We tested different amounts of starting material, different homogenization times, conditions of delipidation and different time and centrifugation speeds for the sucrose cushion step. We were able to develop an optimized protocol to consistently retrieve thousands of nuclei from fat body cells (trophocytes and oenocytes) as well as other associated cell types such as pericardial cells, hemocytes, epidermal cells, and neurons. We do not know the relative efficiency of recovery of the different cell types, because the nuclei isolation and enrichment protocol may introduce some bias, depending on the relative efficiency with which the nuclei from different cells are recovered.

Nuclei quality was verified by light microscopy, with attention to the preservation of DNA content and integrity of nuclear membranes. A manuscript with a detailed description of our nuclei isolation protocol is currently under review in Bio-protocol that we believe will be very useful to the community.

The information regarding the assessment of nuclei quality has been added to the methods section of the revised manuscript: “Nuclei quality was assessed through light microscopy by the evaluation of nuclei membrane integrity.”

6- In Figures 6G-H, authors write « A few hemocytes were also EdU+ in both sugar-fed and blood-fed mosquitoes ». Could the authors please explain how they can identify those cells as hemocytes in the absence of specific staining?

Response:

Point well taken. Morphologically, sessile hemocytes on the mosquito body wall are very small and have no detectable lipid droplets in their cytoplasm. They are typically localized on the surface of fat body lobes, often in close proximity to tracheas. Compared to trophocytes, hemocytes generally have much smaller nuclei, which assists in their identification; however, we acknowledge that this distinction may not always be clear as detection by immunofluorescence with hemocyte-specific antibodies.

To confirm that the observed EdU+ cell were indeed hemocytes, we performed immunofluorescence assays using an antibody against LRR8 (Leucine-rich repeat protein 8), a general hemocyte marker in blood-fed mosquitoes. We observed EdU incorporation in a subset of LRR8+ cells, indicating that blood feeding stimulates DNA replication in hemocytes. A representative image of this co-localization is now included as Supplementary Figure 9. These results are included in the revised results section: “A few cells with small nuclei were observed in both sugar-fed and blood-fed mosquitoes (Figure 6G-H, white arrows). Notably, some of these small-nuclei cells were positive for the hemocyte marker LRR8 (Supplementary Figure 9).”

The methodology to carry out EdU and immunofluorescence staining has been added to the methods section of the revised manuscript:

“5-ethynyl-2'-deoxyuridine (EdU) assay and Immunofluorescence (IFA)

Female *An. gambiae* mosquitoes, 4-5 days post-eclosion, were blood-fed on 3- to 4-week-old female BALB/c uninfected mice (Charles River, Wilmington, MA, USA) for 15 minutes. Prior to feeding, mosquitoes were starved for approximately 16 hours. 5-ethynyl-2'-deoxyuridine (EdU – ThermoFisher Scientific, MA, USA) 3mg/mL 69nL injections were performed at 6 hours post-feeding. Only fully engorged females were selected and maintained on 10% karo syrup following feeding until sample harvesting. Same-age female mosquitoes were also injected with 69nL of EdU 3mg/mL and maintained on ad libitum 10% karo syrup were used as a sugar-fed control. Abdominal fat body tissues were dissected 24 hours post-blood meal or 18 hours post EdU injection, along with their age-matched sugar-fed control mosquitoes. Body walls were dissected in PBS and immediately placed in 4% paraformaldehyde for 30 min at room temperature, briefly transferred to ice-cold 80% ethanol for 3

minutes, and fixed again in 4% paraformaldehyde for 1 hour at room temperature. Tissues were washed three times with PBST 0.1% (1x PBS, 0.1% Triton X-100) for 10 minutes each. EdU was developed as per manufacture instruction with the Click-iT Plus Plus EdU Imaging kit (ThermoFisher Scientific, MA, USA). Briefly, the Click-iT Plus reaction (1x Click-iT reaction buffer, Copper protectant, Alexa Fluor picolyl azide 594nm, 1x Click-iT EdU buffer additive) was prepared and incubated with the sample for 30 minutes at room temperature. Tissues were washed three times with PBST 0.1% for 5 minutes. After this, tissues were blocked for 2 hours at room temperature with blocking buffer (1x PBS, 0.1% Triton X-100, 2% BSA, 0.1% gelatin). Primary antibodies were diluted in blocking buffer: anti-phospho-histone H3 (Ser10) (Millipore Sigma, MA, USA) 1:1,000 or anti-Leucin rich protein 8 (LRR8) (Pacific Immunology, CA, USA) 1:100 and incubated overnight at 4 °C. The next day, body walls were washed three times with blocking buffer for 10 minutes and then incubated with goat anti-rabbit immunoglobulin G (IgG)–Alexa Fluor 555nm (Invitrogen, CA, USA) (1:1,000) diluted in blocking buffer at room temperature for 2 hours. Tissues were washed twice in blocking buffer and then twice more in PBST 0.1%. Finally, tissues were incubated with Hoechst 405nm 1:10,000 and BODIPY 488nm 1:5,000 and Phalloidin 647 nm (1:40) before being placed in ProLong Gold mounting media (Invitrogen, CA, USA).”

Minor comments

7- Given the context that "transcriptional differences between trophocyte clusters are more subtle and unique markers cannot be used to identify specific subpopulations," the authors might consider using trajectory or velocity analysis, to explore whether these clusters represent specific subtypes or continuous states.

Response:

Thank you for suggesting the trajectory and RNA velocity analyses for our dataset.

We performed trajectory analysis using multiple tools, including Monocle 3, Slingshot, and TSCAN, both with and without subsetting the trophocyte clusters. However, in all cases, the cells did not appear to represent continuous states in any of the analyzed samples.

RNA velocity analysis was also not informative for distinguishing whether the clusters represent subtypes or continuous states. However, it revealed dramatic differences in the overall level of unspliced mRNAs, reflecting the dramatic differences in gene expression between sugar-fed and blood-fed samples. When the transcriptional activity of a cell is very high, unspliced mRNAs accumulate in the nucleus. In some cells in clusters T1, T2 and T3 from sugar fed females are transcriptionally active, to maintain the basal function of fat body trophocytes as providing lipids as an energy source, synthesizing the major proteins and lipoproteins circulating in the hemolymph and maintaining a basal level of antimicrobial peptides. [see figure below, now Fig. S6 in the revised manuscript, in which dark blue indicates cells with very low levels of unspliced mRNAs (low transcriptional activity) and dark red indicates high levels of unspliced mRNAs (high transcriptional activity)].

In blood-fed females, the number of cells in clusters T1 decrease dramatically (92%), with a less extreme reduction in clusters T2 (46%) and T3 (38%). The transcriptional activity in cluster T3 also is greatly reduced. A simultaneous increase in the number of cells in cluster T5 is observed, which have extremely high levels of unspliced mRNAs, reflecting very high transcriptional activity of genes involved in oogenesis, such as vitellogenin, which undergoes a 4,000-fold increase in mRNA levels. High transcriptional activity was also observed in the Hm1 hemocyte cluster in response to blood feeding.

Altogether, these analyses reinforce our conclusion that blood feeding induces extensive transcriptional reprogramming, characterized by a shift from cluster T1 to cluster T5, and a strong transcriptional activation in Hm1 hemocytes.

RNA velocity analysis is now included as Supplementary figure 6 and a description of the results was included: “RNA velocity analysis of blood-fed samples indicated enhanced transcriptional dynamics within cluster T5, evidenced by an enrichment of unspliced transcripts and a predominance of positive velocity vectors (Supplementary Figure 6).”

RNA velocity analysis was also added to the methods section:

“RNA velocity analysis

RNA velocity analysis was performed by first generating spliced and unspliced transcript counts from 10x Cell Ranger outputs using `velocyto v0.17.15` [50]. Velocities were then computed and projected onto the Seurat-derived UMAP embeddings using `scVelo v0.2.5` [51] and its dynamical model. Data were preprocessed, filtered, and normalized with “`pp.filter_and_normalize`” (`min_shared_counts=30`, `n_top_genes=2000`), followed by computation of first and second moments using “`pp.moments`” (`n_pcs=30`, `n_neighbors=30`). Finally, “`scv.tl.recover_dynamics`”, “`scv.tl.velocity`”, “`scv.tl.velocity_graph`”, and “`scv.tl.recover_latent_time`” were used to model transcriptional dynamics, estimate RNA velocities, construct directed cell-state transition graphs, and order cells along a latent developmental timeline.”

8- The authors should consider employing differential expression analysis by using pseudo-bulk methods, such as DESeq2 or edgeR, on the single-nucleus RNA-seq data. This approach could help identify the same upregulated genes observed in bulk RNA profiles under conditions of Plasmodium infection, bacterial challenge, or blood feeding.

Response:

We appreciate the reviewer's suggestion to perform pseudo-bulk differential expression analysis using DESeq2 or edgeR. However, our single-nucleus RNA-seq dataset includes one biological replicate per condition, which prevents robust estimation of biological variability required for pseudo-bulk methods. We therefore compared mRNA expression in thousands of nuclei from individual cells and complemented our analysis using bulk RNA-seq from samples obtained in triplicate. This allowed us to confirm and quantitate changes in gene expression in RNA isolated from the whole cells (nuclei and cytoplasm).

9- In Figure 1, panel A illustrates an overview of the experimental workflow, including both snRNA-seq and bulk RNA-seq. However, the rest of Figure 1 only presents results from snRNA-seq. It would be beneficial to include at least a general result from the bulk RNA-seq to align with what is indicated in panel A. In the legend for panel C, the statement "3 representative marker genes per cluster" is inaccurate for T2, which only includes 2 markers.

Response:

Point well taken, we have separated the experimental workflow diagrams for sn-RNAseq and Bulk-RNAseq, they now appear as Fig. 1A and Fig. 3A in the revised manuscript. We have also included more information on the bulk RNA-seq results in Supplementary figures 5 and 8 and corrected the figure legend of panel 1C to reflect the number of genes markers for the T2 cluster in the dot plot.

10- While the authors indicate that Galactokinase (GALK) serves as a marker for all trophocyte clusters (line 122-124), it does not appear highly expressed in T5 (Figure 1C, Figure 2B). This observation weakens the claim that GALK is a universal marker across all trophocyte profiles, regardless of blood feeding status. This, this claim could be moderated.

Response:

We have modified the claim to state that Galactokinase (GALK) is a marker of sugar-fed trophocytes cluster. We have changed the sentence to: "Five distinct trophocyte transcriptional profiles were observed, with clusters T1, T2, T3 and T4 present in sugar-fed samples and cluster T5 present only in blood-fed samples. Galactokinase (GALK - AGAP002914) is a general marker gene expressed in sugar-fed trophocyte clusters (Figure 1C, Figure 2B)."

11- Fatty acid synthase (FAS) has multiple isoforms in some mosquito species. Could the authors clarify the specific FAS gene or isoforms measured in *Anopheles gambiae*? Are certain FAS isoforms uniquely enriched in oenocytes (line 138-139), or are they distributed across various cell types, including

trophocytes? Understanding the specialization of FAS in oenocytes would provide valuable insights.

Response:

In *Anopheles gambiae* PEST genome (AgamP4.14, v59), there are three annotated as fatty acid synthase (FAS) genes— AGAP001899, AGAP008468, and AGAP009176. The FAS genes AGAP001899 and AGAP008468 are both markers of oenocytes and are upregulated in primed mosquitoes based on the bulk RNA-seq data. The FAS gene AGAP009176 is not a specific marker of the clusters found in our snRNA-seq dataset and it was not regulated in primed mosquitoes. We selected the gene AGAP001899 as example to validate our results with qPCR and RNA FISH (FAS1899 probe), and we observed an increase in gene expression in primed mosquitoes and a specific staining of oenocytes with the FAS1899 probe. The feature plots of the three FAS genes in our dataset are shown in the figure below, with AGAP001899 and AGAP008468 highly expressed in oenocytes (dark orange), while the AGAP009176 gene is highly expressed in many other clusters.

12- The results of the bulk transcriptomic analysis after bacterial challenge (lines 208–222), including the regulation of antimicrobial, immune, and metabolic pathways, are mentioned only in a supplementary table. The authors might consider adding a main or supplementary figure to enhance the impact and accessibility of these findings. In addition, the strong transcriptional response to blood feeding, (lines 286-310) is currently presented only in supplementary tables, which diminishes their impact. To enhance clarity and strengthen these findings, the authors might consider adding them in a main or supplementary figure.

Response:

We thank the reviewer for the suggestion. To enhance the clarity and impact of our bulk transcriptomic results, we have added two figures (Figs. S5 and S8) combining volcano plots and heatmaps to display examples of differentially expressed genes after bacterial challenge or blood feeding, highlighting the magnitude and significance of transcriptional changes. These panels provide a concise and accessible

representation of the bulk RNA-seq results, which were previously only included in supplementary tables (Supplementary Figures 5 and 8).

13- In Figure 2A, the authors might add another item in the legend to specify what the grey boxes behind the central and dorsal tissues represent.

Response:

Thank you for your observation. We included the description of the gray boxes to the Figure 2A legend. It now reads: "Gray boxes represent three segments of abdominal wall."

14- Would the author be able to clarify which study or facts supports the following claim : "lipids interfere with cell dissociation and 10X single-cell partitioning and Gel Bead-in-Emulsion (GEM) formation?"

Response:

We thank the reviewer for the request to clarify the statement regarding lipids. While we did not locate a publication that explicitly states "lipids interfere with GEM formation in the 10x workflow," there is supporting documentation from the 10x Genomics company indicating that high-lipid tissues (e.g., adipose) present challenges in pelleting and single-cell suspension preparation (10x Genomics "Single Cell Flex Tested Tissues" support page) and that non-ideal suspension conditions (e.g., debris, non-singlet cells) may lead to microfluidic clogs or wetting failures in GEM generation (10x Genomics knowledge base). In our experience, incomplete dissociation or the presence of lipid aggregates can also lead to microfluidic channel clogging. To mitigate these issues, we included the delipidation step in the nuclei isolation protocol, which allowed reliable GEM formation and consistent single-nucleus capture.

15- The authors might want to include a definition of endoreplication in their discussion.

Response:

The definition of endoreplication was included in the discussion section. It now reads: "Endoreplication is a type of cell cycle where there is DNA replication without cell division, leading to polyploid cells."

Reviewer #2 (Remarks to the Author):

Response:

Thank you for the initiative to provide opportunity to early career researchers.

Reviewer #3 (Remarks to the Author)

In this article the authors present a well-executed set of experiments at single-nucleus resolution to characterise the complexity of the mosquito fat body—an organ central to both female reproductive development and innate immunity. The work is relatively descriptive in nature, as is often the case with this approach, but the level of cellular and molecular complexity revealed is valuable and will provide an important reference point for the field. It is therefore likely to be frequently cited.

The introduction is exemplary in its clarity. However, both I and a more junior colleague who helped me review the manuscript felt that some important detail was missing from the description of the experimental set-up used to examine the effects of immune priming. This is a field in which the authors have made many of the seminal contributions—indeed, they effectively “wrote the book” on the subject—so the rationale deserves to be laid out more fully. The hypotheses underpinning these experiments could be stated more explicitly, and fewer assumptions should be made about the reader’s familiarity with why particular time points, bacterial delivery methods, and bacteria–*Plasmodium* combinations were chosen.

A similar point applies to aspects of the discussion concerning transcript localisation and endoduplication. The interpretations themselves are reasonable, but the arguments would be clearer with more explicit rationale and fewer assumptions about prior knowledge. A little more empathy for the reader in these explanations would substantially strengthen the manuscript.

More detailed comments and queries are provided below:

In the results section there needs to be an explanation of why they looked at the conditions they did. It would be helpful to add a justification of why both single nucleus and bulk sequencing was used (i.e. what different results were expected from each one). Also on the experimental rationale, the manuscript would be easier to understand if the authors add an explanation of why systemic bacterial challenge by injection into the hemolymph is an important and relevant immune challenge to mosquito biology, including explaining the choice of bacteria used in the challenge (perhaps representatives of gram positive/ negative?)

Response:

Thank you for the helpful suggestion to expand the explanation of our experimental design. We have revised the results section to clarify the rationale behind the selected conditions and the use of both bulk and single-nucleus RNA sequencing.

We added a detailed explanation describing the complementary nature of bulk and snRNA-seq and why both approaches were necessary for this study: “In this study, bulk and snRNA-seq were used to identify both broad transcriptome changes, as well as cell-specific responses of *Anopheles gambiae* adult female fat body trophocytes to blood-feeding, bacterial challenge and immune priming (Figure 1A). Bulk RNA-seq was used to capture broad transcriptional changes with high depth coverage, allowing us to quantify global differences in gene expression across different physiological and immune conditions. In contrast, snRNA-seq enabled the resolution of cell-type–specific transcriptional responses and the identification of different trophocytes subpopulations. Using both methods allowed us to distinguish tissue-wide shifts

from changes restricted to specific fat body cell types, providing a more comprehensive understanding of how these stimuli reshape fat body biology.”

We expanded the description of the physiological and immune conditions analyzed in this study, explaining their biological relevance to fat body function and mosquito immunity. The text in the revised manuscript now reads: “Due to the importance of the fat body tissue on reproduction and immunity samples were collected from the abdominal wall of healthy *An. gambiae* females fed on sugar or blood, as well as from females systemically challenged with bacteria or primed by infecting them with *Plasmodium berghei* (Figure 1A). Samples from females challenged systemically with a mixture of *Escherichia coli* and *Micrococcus luteus* were analyzed 4 hours post-injection, while the priming response was evaluated six days after females were fed on *P. berghei*-infected mice. The response of fat body cells to blood feeding was investigated 24 hours after ingestion of a blood meal. The selected conditions represent the major physiological and immune challenges experienced by *An. gambiae* adult females: nutrient acquisition (sugar feeding), reproductive activation (blood feeding), acute antibacterial defense (systemic bacterial challenge), and long-term immune modulation (*Plasmodium*-mediated priming). Together, these treatments capture a broad spectrum of stimuli that shape fat body function in nutrition and vector–pathogen interactions.”

We included additional details explaining the experimental design of the systemic infection and its importance for understanding fat body immune responses, as follows: “The fat body plays an important role in the mosquito systemic immune response by producing antimicrobial peptides and phenoloxidase enzymes that are secreted into the hemolymph and limit bacterial proliferation and mediate pathogen melanization, respectively. However, cell type-specific responses remain poorly understood. The transcriptional response of the abdominal body wall to bacterial infection was explored by challenging adult females with a systemic injection of a mixture of a Gram-negative (*E. coli*) and a Gram-positive bacteria (*M. luteus*), or a sterile PBS control, and harvesting the abdominal body wall 4 hours later. This approach ensured the direct delivery of a standardized bacterial load into the hemolymph, triggering a robust and physiologically relevant immune response. The combination of Gram-positive and Gram-negative bacteria provided a broad immune stimulus, ensuring the activation of the major immune signaling pathways.”

Several sections of the results presented interesting but incomplete stories. Adding either further experiments or at least fuller interpretation of the observations with clearer recommendations for future work would greatly improve the clarity and interest of the manuscript to a wider audience. These included:

1. Does the trade-off between immunity and metabolism mentioned on p18 of the MS (lines 393-397) extend to measurable impacts on reproduction (as explored for *Drosophila* in Gupta et al 2022 DOI: 10.1186/s12915-022-01328-w)? Importantly, how does bacterial or *Plasmodium* challenge impact the DNA replication or expression of yolk protein genes in trophocytes?

Response:

Thank you for citing Gupta et al. (2022, BMC Biology; DOI:10.1186/s12915-022-01328-w). We agree that their work is highly relevant to the trade-off concept we discuss. In their *Drosophila* study, they used single-nucleus sequencing to define subpopulations of fat body cells and showed that, following bacterial infection, all subpopulations mount immune responses. Crucially, in mated (reproductively active) females, they saw ER stress and an impaired capacity to upregulate translation, including antimicrobial peptide synthesis, which they linked to reduced survival. By temporarily inhibiting translation before infection, they were able to restore immune function and survival, suggesting that protein synthesis capacity is a major bottleneck when fat body must support both reproduction and immunity.

In our study, we did not observe transcriptional changes in mRNA expression of yolk protein genes or in DNA-replication-associated genes in response to bacterial challenge or *Plasmodium* priming under the experimental conditions we used. However, it is possible that these challenges had an indirect effect on reproduction, since mosquitoes must allocate energetic resources between reproductive investment and immune activation, and such physiological trade-offs may not necessarily manifest at the transcriptomic level under the conditions tested here.

We agree that future experiments—for example, measuring translation capacity, ER stress markers, or reproductive output (egg production) in challenged versus unchallenged mosquitoes—could test whether a trade-off similar to that seen in *Drosophila* exist in *Anopheles*. We agree with the reviewer that these are excellent questions, and it would be interesting to address them in future research, but they are beyond the scope of the current manuscript.

2. Vitellogenin mRNA localisation. A possible parallel with past observations in *Aedes* trophocytes entering the vitellogenic phase is mentioned (lines 412-420), with ribosomes relocating and plasma membrane changes. Figure 6 shows a clear localized expression of Vg mRNA in the trophocytes at the hemolymph side using in situ hybridisation. However, given that the single cell results were based on single nuclei, one might expect that Vg mRNA would additionally be enriched in the nuclei of these cells compared to the other cell types, since transcripts outside of the nucleus should have been removed during nucleus preparation and their data processing steps to remove ambient mRNA. In figure 6 Vg mRNA cannot be seen in trophocyte nuclei. It should be explained why this is. Is this because the level in the nuclei is low compared to the strong cytoplasmic accumulation, or is it due to contamination of the sequenced nuclei with non-nuclear mRNA? Furthermore, there was a general issue with all in-situ microscopy presented that there was no marker for the cell membrane or sub membrane cytoskeleton, making it impossible to interpret where the cell margins were. Adding a cell outlining marker would really aid interpretation of these images for the reader and make the cell specific markers much more convincing.

Response:

The reviewer brings some interesting observations. Nuclear mRNA typically represents about 10% of cytoplasmic mRNA, although this proportion can vary across species and cell types. Additionally, certain mRNAs can be localized to specific subcellular regions, resulting in higher concentrations in those areas.

We believe this is the reason why all the probes for different genes had a strong cytoplasmic signal while the nuclei signal could not be readily detected.

Based on the bulk-RNA sequencing, which includes both the nuclei and cytoplasm, Vg mRNA expression increases more than 4,000-fold in blood-fed mosquitoes, resulting in a very strong signal in the RNA FISH experiments. The lack of detection of Vg RNA in the nuclei in *in situ* hybridizations is due to a much higher concentration of mRNA in a very narrow region of the cytoplasm, on the side of the trophocytes facing the hemolymph, and we need to reduce the potency of the laser to not saturate the signal coming from a very narrow area in the cytoplasm. We took extensive steps to clean the samples and applied decontamination strategies to minimize ambient RNA contamination; however, a minor level of contamination is possible.

We agree that the inclusion of a membrane or cytoskeleton marker would have improved interpretation of the RNA FISH images by providing clearer visualization of cell boundaries. We tested several membrane dyes and cytoskeletal markers during protocol optimization; however, none were compatible with our RNA FISH conditions, as they either interfered with probe hybridization or were lost during the denaturation and wash steps. For this reason, we used BODIPY, which reliably stains neutral lipids and proved especially useful for distinguishing trophocytes from other fat body cell types, as their cytoplasm is filled with lipid vesicles.

3. More explanation of how cells were counted is needed. A fuller explanation of the rationale for counting cell types based on single nucleus sequencing and the error of these measurements (where replicate experiments allow this to be estimated) should be added to the methods section. This is needed to validate the many interesting observations on the changes in the proportions and numbers of cell typed during different treatments. For example, the story regarding circulating versus static granulocytes. In the introduction, lines 61-67 mention that immune priming in *An. gambiae* is mediated by release of hemocyte differentiation factor when microbiota contacts the midgut epithelia, and that this triggers an increase in the proportion of circulating granulocytes. Theoretically a changing proportion between circulating and fat-body associated granulocytes could be achieved either by release of fat body granulocytes to the haemolymph (resulting in fewer fat body granulocytes but more circulating), or by cell division which would boost the circulating population leaving the fat-body associated population at the same size. These alternative hypotheses could have been investigated using their combined single nucleus followed by *in situ* hybridisation methods. In their investigation of the response of fat body associated granulocytes to *P. berghei* challenge there was a clear transcriptional change but crucially, they state that they did not observe a significant difference in the number of body wall-associated granulocytes (results lines 181-182). This leaves the reader wondering how they measured the number of granulocytes. Was this based on the relative proportion of nuclei with a granulocyte expression profile compared to other fat body cell types in the single nuclei sequencing? If this is the case, what is the quality control for this – how do we know that the expression and physiological changes associated with the *P. berghei* challenge (or any of the other treatments used in the manuscript) did not influence the recovery rate of nuclei from different cell types differently and bias the count? Can this observation be confirmed using HCR-FISH and counting fat body associated granulocytes based on cell-type specific markers in naïve vs challenged

mosquitoes? Would the final interpretation of this result be that the boost to circulating granulocytes is not by release of fat body associated granulocytes? This issue of how cells of each type were counted and how confident we can be in the counts/ relative proportions going on single nucleus clustering alone is also relevant for the stated observations about trophocytes (lines 263-265), granulocytes following bacterial challenge (lines 356-360) and oenocytes (p369-371).

Response:

We appreciate the reviewer's thoughtful comments regarding cell quantification and the interpretation of changes in cell proportions. We would like to clarify that we did not perform absolute cell counts in this study. Our analysis relied on the number of nuclei recovered in each cluster in the snRNA-seq dataset, and these values were used only to describe the relative representation of transcriptional states rather than to infer absolute differences in cell numbers across conditions. We agree entirely that nuclei recovery can be influenced by technical factors, including differential nuclear stability or accessibility across cell types and treatments; therefore, we did not interpret shifts in cluster representation as quantitative biological counts. However, we think it is important to compare and document the relative abundance of different cell types between control and experimental samples that were processed side by side to prevent differences due to batch effects.

Our goal in presenting these observations was to highlight transcriptional changes and raise biologically meaningful hypotheses—for example, regarding the relationship between sessile and circulating granulocytes—rather than to conclude mechanistically how these populations change *in vivo*. As the reviewer points out, distinguishing between granulocyte recruitment, release, or proliferation would require direct quantitative approaches such as FISH-based cell counting, imaging of labeled hemocytes, or lineage-tracking methods. Quantifying hemocytes embedded within the fat body is technically challenging, and we did not attempt it here.

We have clarified in the methods section that the estimated cell proportions were based on the number of captured nuclei: "The estimated relative cell abundance provided in Supplementary Table 2 represents the relative proportion (%) of nuclei from each cluster relative to the total nuclei captured in each sample."

Based on the reviewer's comments, we modified the text regarding the cell proportion changes between conditions: "Although we observed transcriptional differences between granulocytes from naïve and primed females, we did not observe a difference in the proportion of captured nuclei from body wall-associated granulocytes in response to immune priming." and "We observed a doubling of the number of fat body-associated granulocytes following a bacterial challenge in snRNA-seq, suggesting either proliferation of resident cells and/or recruitment from circulation."

We agree that the questions raised—particularly concerning the balance between circulating and fat-body-associated granulocytes and the interpretation of trophocyte and oenocyte proportions—are important and remain open in the field. We are actively pursuing these issues in ongoing and future work.

4. The story regarding endoreplication in trophocytes was very interesting but seemed to finish before reaching a firm conclusion. They observed EdU-positive but p_{H3} negative trophocytes which would be consistent with DNA replication without mitosis. Why did they not attempt to look at the ploidy in these cells? This could be done using for example FACS or even just measuring the brightness of nucleic acid stains such as Hoechst using microscopy. Similarly, given that they applied a filter to removed doublets from single nucleus data, can they be sure they did not accidentally filter out trophocytes with a higher ploidy – how did the various QC metrics differ between potential doublets and endoreplicated nuclei? Furthermore, in this study they only investigated the response of the fat body to the first blood meal. If the endoreplication occurs after the first blood feeding to boost transcription, this raises questions as to what happens to ploidy after 6 days when the transcriptional profile returns to pre-blood fed patterns, particularly in their cluster “T1” – are all these extra DNA copies kept? Or perhaps could the “T1” cluster 6 days after blood feeding be new cells and not the same T5 endoreplicated cells returning to pre-blood fed transcriptional patterns? If the cells in T1 are the T5 cells and ploidy remains at this boosted level, does that mean that after subsequent bloodmeals the transcriptional shift would be different since it should no longer require genes involved in DNA replication since the endoreplication has already happened?

Response:

Thank you for your questions and for suggesting experiments for future studies.

We performed the EdU and phospho-histone H3–positive (PH3) assays to determine if the drastic changes in the number of cells in clusters T1 and T5 observed 24h after ingesting a blood meal could be due to cell replication giving rise to a new cell population (T5). The EdU staining pattern in the blood-fed samples showed that most trophocytes are EdU-positive, with some heterogeneity in the intensity, and negative for PH3 staining. This indicates that most trophocyte from different subpopulations undergo some level of DNA replication without cell division (endoreplication). These findings support the hypothesis that the T1 cluster undergoes a dramatic transcriptional change, with endoreplication and very high expression of specific genes, such as *Vg*, giving rise to a very different transcriptional profile that the software defines as a new (T5) cluster. This is a transient response, as most cells with the T5 transcriptional profile disappear when oogenesis is completed (6 days post-feeding) and return to the T1 transcriptional state.

We did not assess the ploidy of trophocytes in this study because a previous publication in *Aedes aegypti* have already reported an increase in trophocyte ploidy following a blood meal [3]. We agree that it would be very interesting to carry out *in vivo* EdU staining and isolate nuclei and carry out FACS analysis with both EdU and Dapi staining, to determine the level of ploidy and how it changes after the first or second blood meals. Those are interesting future experiments.

We analyze the distribution of the number of genes and UMIs in the nuclei from each sample individually and use a software to identify potential doublets. However, no doublet nuclei were detected in any of the samples. Doublets are more problematic when working with dissociated tissues, as sometimes the dissociation is incomplete or when the membrane of two dissociated cells “stick” to each other. With the buffer conditions we used, we never observed nuclei doublets, in agreement with the

bioinformatic analysis. Indeed, we observed that a substantial number of nuclei from the blood-fed samples displayed higher numbers of detected features and UMIs compared to sugar-fed controls, but none of those cells were removed from the analysis.

We agree that it would be interesting to examine the trophocyte ploidy at 6 days post-feeding, to determine if it persists and, if so, whether it would further increase after a second blood meal. If the increase in ploidy persists, it may enhance vitellogenesis by supporting higher expression of yolk protein genes. To our knowledge, no studies have reported how trophocyte ploidy changes following a second blood meal. We appreciate the interesting suggestions, but they are beyond the scope of this manuscript.

5. On the report of the result that hemocytes are the second most abundant cell type (lines 124-125) "Surprisingly, hemocytes (Hm1, Hm2) are the second most abundant cell type, representing 7.4% of the cells (Supplementary Table 2)", it needs to be explained why this is surprising?

Response:

Thank you for this comment. Most of the quantitation of hemocytes is based on circulating hemocytes. We surprised, because although we have work with hemocytes for many years, we were not aware that there were so many hemocytes associated with the fat body tissue. To be more objective, we have modified the text, removing the observation that we are surprised, and now it reads: "Hemocytes (Hm1, Hm2) are the second most abundant cell type, representing 7.4% of the cells (Supplementary Table 2)."

6. The question of whether cells are "resident" or "associated" with the fat body does raise the question of whether they could move in and out. This is particularly relevant to the topic of the parallels between insect fat body and the vertebrate immune system (e.g. the potential role of T4 trophocytes in immune surveillance discussed lines 391-393), a topic that is of strong interest to the general reader. They postulate in lines 356-357 "We observed a doubling of the number of fat body-associated granulocytes in response to bacterial challenge, suggesting either proliferation of resident cells and/or recruitment from circulation". Although directly testing the alternative hypotheses of cell movement vs cell proliferation might be beyond the scope of this study, this question should be better highlighted in the discussion with a recommendation as to what sort of experimental approach could definitively answer this question in future studies (for example tracking of cells over time).

Response:

We are very interested in exploring these aspects in future studies comparing sessile and circulating hemocytes. To our knowledge, there are currently no reports describing a hematopoietic organ in adult mosquitoes, which raises the intriguing question of how hemocyte populations are replenished. Our group previously identified a population of circulating, dividing granulocytes, and in the present study, we also observed evidence of cell division (PH3-positive cells) among sessile hemocytes.

The mechanisms regulating the dynamics between tissue-associated (sessile) and freely circulating hemocytes remain poorly understood. To address these questions, we plan to perform single-cell RNA sequencing to compare sessile and circulating hemocytes in the same mosquitoes and investigate their

response to different stimuli, including immune priming and bacterial infection, using a combination of IFA and RNA FISH microscopy.

We have modified the Discussion section to include a proposed experimental approach addressing these points: “Future studies to explore the relative contribution of proliferation of fat body-associated hemocytes and recruitment of hemocytes circulating in the hemolymph will require longitudinal tracking and quantification of hemocyte populations in the same experimental samples, using a combination of single-cell RNA sequencing and high-resolution tissue imaging.”

Further discussion of how the Anopheles fat body is similar or different from other insects would have improved the discussion section. This seems part of the stated aim of addressing a knowledge gap in Anopheline mosquitoes about the cellular composition and physiology of the fat body. The discussion section alludes to studies of the fat body cellular composition in other organisms including *Drosophila melanogaster* and *Bombyx mori*, and yet there is no discussion of the similarities and differences between what they observed in *Anopheles gambiae* fat body single nucleus sequencing and these earlier works.

Response:

Thank you for the opportunity to expand on the discussion of the comparative morphology of the fat body between species and explain the differences between the previous studies on *Drosophila*, *Bombyx mori*, and *Aedes aegypti*. We added the following text to the discussion section, as suggested by the reviewer: “*Drosophila melanogaster* and *Bombyx mori* are non-hematophagous insects that feed continuously, and that fundamental dietary difference shapes fat body biology in ways that contrast with mosquitoes. In *Drosophila*, the fat body forms diffuse sheet-like layers with compact segmental oenocyte clusters [31], whereas *B. mori* possesses large, well-defined lobes and abundant oenocytes [32]. In mosquitoes such as *Ae. aegypti* and *An. gambiae*, however, the requirement of a blood meal for egg development drives drastic and transient transcriptional and physiological changes in fat body trophocytes to sustain vitellogenesis [7, 12]. The mosquito fat body is organized into distinct lobes with more dispersed oenocyte clusters [7]. In our study, we identified different trophocyte subpopulations, as well as oenocytes and other fat body-associated cell types, including hemocytes, epidermal cells and neuronal cells, in agreement with previous single cell studies in insects [28-30].”

There are a few technical considerations. As stated by my colleague above, the manuscript appears to be generally technically sound, but with the rationale of experiments, some technical issues warranted a clearer explanation. In the discussion the authors claim to have overcome technical barriers to nucleus isolation (line 330, 432). They give no details of these technical barriers and don't show any quality control of the nuclei isolated prior to sequencing (for example microscopy to show nuclei morphology). They seem to have followed initially a protocol similar to what was used in Gupta and Lazzaro 2022 (<https://doi.org/10.1080/19336934.2021.1978776>) followed by using the Sigma “Nuclei Pure Prep” standard protocol. Were other options for nuclei isolation tried and worked less well? As a minimum, images of the isolated nuclei for each treatment and yield as expressed as amount of nuclei/number of tissues dissected should be shown. It could also be acknowledged that *Anopheles* oenocytes have been isolated in the past (Grigoraki et al 2020, 10.7554/eLife.58019), although this

was with the help of a transgenic GFP expressing line whereas the method developed in the current manuscript are applicable to all genotypes.

Response:

We have submitted a detailed method manuscript to Bio-protocol where we provide the nuclei images and estimated yield of nuclei recovery, and it's currently under review. With this protocol we were able to obtain high-quality nuclei, yielding approximately 10^6 nuclei from 25 body wall tissues. We added to the discussion the technical challenges faced on isolating single cell from the fat body tissue that now reads: "Trophocytes are large cells with the cytoplasm filled with lipid droplets, making them fragile to high-quality single-cell isolation. Our optimized protocol involves a delipidation step, because high lipid content in the sample can lead to nuclei aggregation and interference with 10X single-cell partitioning and Gel Bead-in-Emulsion (GEM) formation. In our experience, we were unable to obtain high-quality intact fat body cells using enzymatic methods of tissue dissociation."

We tested the protocol from Grigoraki I. 2020 that used trypsin to dissociate the tissue. In our experience this enzymatic treatment was harsh on the trophocytes, leading to poor-quality isolated cells and RNA degradation and very few oenocytes were dissociated. Our goal was to characterize as many cell types associated with the fat body as possible. In our experience, oenocytes are much more resistant to the enzymatic treatments compared to trophocytes.

The number of biological replicates used for the single cell work was very low. Supplementary figure 1G suggests 2* "naïve", 2* "challenge", and only one each of other conditions (although "control" might be the same as one of them?). This low level of replication is likely due to the high cost of undertaking the single cell work but means that any important observations such as on the relative numbers of cells in each cluster need to be confirmed by another method for statistical robustness.

Response:

Response: Thank you for the comment. Single-cell sequencing is usually carried out with a single sample, because the transcriptome from each cell is considered a replicate of how a tissue responds to a stimulus. We performed two biological replicates for the naïve and primed samples and one technical replicate of the sugar-fed, blood-fed, PBS injected control and bacterial injected samples.

Although we agree that proportion of captured cells are not the best method to estimate cell count and that capturing bias might occur during library preparation, the proportion of cells in each cluster were similar between different conditions. We also processed the control and experimental samples for each treatment side by side, on the same day to prevent any difference due to batch effects. We also carried out bulk-RNA sequencing in triplicates, which would reflect the transcriptional responses of the whole cell (nuclei and cytoplasm). The results of the snRNAseq are in agreement with the Bulk-sequencing analysis.

We mentioned the cell differences as support information consistent with the observed transcriptional changes, and highlighted only the dramatic shift in trophocyte populations, T1 to T5, between sugar-fed

and blood-fed samples. The hypothesis raised regarding the shifts in cell populations such as granulocytes are very interesting but outside the scope of this manuscript.

On the analysis of the single nucleus sequencing data, lines 102-103 state that the “cutoffs for the number of counts and features, as well as the percentage of mitochondrial and ribosomal content, were set for each sample (Supplementary Figure 1)”. However, although supplementary figure 1 shows the spread of the data it does not actually indicate what cutoffs were used or how the decision to set these cutoffs was made. This should be made clear.

Response: Thank you for the suggestion. The cutoffs were described in the methods section: “Quality control and cell filtering were performed using the Seurat R package (version 4.2.1). For each sample, nuclei were retained based on sample-specific thresholds for the number of detected genes (Features), the percentage of mitochondrial gene expression, and the percentage of ribosomal gene expression, to remove low-quality nuclei and potential doublets. Cells were included if they expressed between 200 and 2000 genes for most samples (Naïve 1, Naïve 2, Prime 1, Prime 2, and Blood-fed), with adjusted upper bounds for PBS-control, bacteria-injected (2500), and Sugar-fed samples (1500). To minimize the inclusion of damaged or stressed nuclei, cells with a mitochondria gene percentage exceeding 2.5% (Naïve 1 and Prime 1), 1.5% (Naïve 2 and Prime 2), 7% (PBS-control), 5% (Bacteria-injected), or 10% (Sugar-fed and Blood-fed) were excluded. Additionally, a ribosomal gene expression threshold was applied, with upper limits ranging from 10% to 25% across samples to account for potential variation in ribosomal content between conditions.”

The selection of cluster specific markers seemed to have a very low bar in terms of the fold change (Avg log₂FC > 0.5) – just 1.41 times more expression in the enriched clusters? The authors should explain how this cut-off was decided on. In supplementary table 1 where these markers are described, the proportion of cells with detectable expression was also rather low for some marker genes.

Response:

Thank you for your consideration about the cutoffs. The avg log₂FC > 0.5 threshold (≈1.4-fold) is a standard and widely used cutoff in single-cell RNA-seq analyses, including Seurat-based workflows. It is intentionally modest because single-nucleus data exhibit higher dropout rates and smaller dynamic range than whole-cell scRNA-seq. Stronger thresholds (e.g., >1 or >1.5 log₂FC) would eliminate many biologically meaningful but sparsely detected nuclear transcripts. We therefore used a conservative fold-change cutoff combined with specificity criteria (proportion of expressing cells, adjusted p-value, and differential expression compared to all other clusters) to ensure that markers reflected consistent enrichment despite nucleus-based technical dropout. The lower proportion of detectable expression for some markers is expected for transcription factors and low-abundance genes in snRNA-seq datasets and is not unusual.

Lines 531-533 mention that “The gene annotation (gtf) file was filtered to retain protein-coding genes”. Could valuable data on the expression of non-protein coding genes such as long noncoding RNAs have been discarded at this point and how would its inclusion have affected the clustering?

Response: Thank you for your comment. Filtering the annotation file to retain protein-coding genes is a common step in snRNA-seq processing pipelines, particularly for species with incomplete or inconsistently annotated non-coding transcriptomes, such as *Anopheles gambiae*. While this approach removes long noncoding RNAs (lncRNAs) and other non-protein-coding features, the impact on clustering is expected to be minimal for two reasons. First, most lncRNAs in mosquitoes remain poorly annotated, fragmented, or lacking validated transcript boundaries, which complicates quantification and can introduce noise rather than meaningful signal. Second, clustering in single-nucleus datasets is overwhelmingly driven by the highly expressed and well-annotated protein-coding transcriptome; low-abundance or sparsely detected lncRNAs typically contribute little to the principal components that define cluster structure. Thus, while lncRNAs may have biological importance, their exclusion is unlikely to have altered the clustering results, and including poorly annotated non-coding features could have increased technical noise without adding interpretable information. In a future project, we are planning to use the Bulk-RNA sequencing data to attempt to map all the reads to the genome and annotate new transcripts, including lncRNAs, for these body wall samples. The assembly and annotation will be a time-consuming project, but once completed, it will allow us to re-analyze our snRNA-seq data to investigate the potential cell-specific expression and transcriptional responses of lnc-RNAs.

For the ambient RNA correction (lines 537-538 “Correction was performed using the decontX algorithm from the celda package [48]” it should be specified if the same model was applied across all samples or whether this was modelled individually in each sample.

Response: Thank you for the suggestion. The decontX model was applied to each sample before integration. Methods section was modified to include this information: “Correction was performed in each sample using the decontX algorithm from the celda package [49], which models and removes contamination from ambient background RNA in snRNA-seq data.”

Reviewer #4 (Remarks to the Author):

de Carvalho and coworkers present a single nuclei RNAseq data set of *Anopheles gambiae* fat body. They show a variety of cell types, and notably assign five trophocyte clusters. They show some cells responding to bacterial challenge and, not surprisingly see a strong upregulation of vitellogenesis-related genes after a blood meal.

The introduction is beautifully written, the experiments are well described. The results show the expected heterogeneity of cell types in this tissue.

This is a well-done study.

Response: Thank you for the positive feedback.

Minor points:

- a scheme of where the different trophocyte types are located within the abdomen would be helpful.

Response: Thank you for the suggestion. The trophocyte subpopulation did not show a particular localization within the abdominal body wall, and we were able to stain all trophocyte populations on the lateral, ventral and dorsal sides of the tissue.

- could the tissue and subcellular locations of some mRNAs you see (especially Vg) be due to technical problems of the in situ hybridization protocol? Just asking questions...

Response: The reviewer raises an important point. We had the same concern. We carefully evaluated the possibility that the observed mRNA localization patterns could result from technical artifacts. The RNA FISH experiments were repeated multiple times, and the subcellular localization of vitellogenin (Vg) was highly reproducible across independent dissections and hybridizations. Furthermore, the probes for several other genes hybridized in the same samples as the Vg probe and stained all the cytoplasm and did not have the localized subcellular localization of Vg. This rules out the possibility of the lack of homogeneous staining of the cytoplasm with the Vg probe was due to a lack of permeabilization. We also obtained the same localization pattern using an alternative probe targeting a different gene that is also highly induced by blood feeding. Taken together, these controls support the conclusion that the observed Vg mRNA localization reflects a genuine biological distribution rather than technical artifacts.

- a longer discussion of how the results of [30] compare to these findings would be appropriate.

Response: Thank you for the suggestion. We have expanded the discussion comparing our findings to previous single cell papers: “*Drosophila melanogaster* and *Bombyx mori* are non-hematophagous insects that feed continuously, and that fundamental dietary difference shapes fat body biology in ways that contrast with mosquitoes. In *Drosophila*, the fat body forms diffuse sheet-like layers with compact segmental oenocyte clusters [31], whereas *B. mori* possesses large, well-defined lobes and abundant oenocytes [32]. In mosquitoes such as *Ae. aegypti* and *An. gambiae*, however, the requirement of a blood meal for egg development drives drastic and transient transcriptional and physiological changes in fat body trophocytes to sustain vitellogenesis [7, 12]. The mosquito fat body is organized into distinct lobes with more dispersed oenocyte clusters [7]. In our study, we identified different trophocyte subpopulations, as well as oenocytes and other fat body-associated cell types, including hemocytes, epidermal cells and neuronal cells, in agreement with previous single cell studies in insects [28-30].”

References

1. Li, M., et al., *Response of the mosquito immune system and symbiotic bacteria to pathogen infection*. Parasit Vectors, 2024. **17**(1): p. 69.
2. Sigle, L.T. and J.F. Hillyer, *Mosquito hemocytes preferentially aggregate and phagocytose pathogens in the periostial regions of the heart that experience the most hemolymph flow*. Dev Comp Immunol, 2016. **55**: p. 90–101.
3. Dittmann, F., P.H. Kogan, and H.H. Hagedorn, *Ploidy Levels and DNA-Synthesis in Fat-Body Cells of the Adult Mosquito, Aedes-Aegypti - the Role of Juvenile-*

Hormone. Archives of Insect Biochemistry and Physiology, 1989. **12**(3): p. 133–143.